# Cause-effect inference through spectral independence in linear dynamical systems: theoretical foundations.

**Michel Besserve**[1,2]                                    BESSERVE@TUE.MPG.DE
**Naji Shajarisales**[1,3]                                         NAJIS@CMU.EDU
**Dominik Janzing**[4,*]                                    JANZIND@AMAZON.DE
**Bernhard Schölkopf**[1]                                        BS@TUE.MPG.DE

*1. Department of Empirical Inference, MPI for Intelligent Systems, Tübingen, Germany.*

*2. Department of Cognitive Physiology, MPI for Biological Cybernetics, Tübingen, Germany.*

*3. Carnegie Mellon University, Pittsburgh, USA.*

*4. Amazon Research Tübingen, Germany.*

*\*. DJ contributed before joining Amazon.*

**Editors:** Bernhard Schölkopf, Caroline Uhler and Kun Zhang

## Abstract

Distinguishing between cause and effect using time series observational data is a major challenge in many scientific fields. A new perspective has been provided based on the principle of *Independence of Causal Mechanisms* (ICM), leading to the *Spectral Independence Criterion* (SIC) for time series causally unidirectionally linked by a linear time-invariant relation. SIC postulates that the power spectral density (PSD) of the cause time series is *uncorrelated* with the squared modulus of the frequency response of the filter generating the effect. Since SIC rests on methods and assumptions in stark contrast with most causal discovery methods for time series, it raises questions regarding what theoretical grounds justify its use. In this paper, we provide answers covering several key aspects. After providing an information theoretic interpretation of SIC, we present an identifiability result that sheds light on the context for which this approach is expected to perform well. We further demonstrate the robustness of SIC to downsampling – an obstacle that can spoil Granger-based inference. Finally, an invariance perspective allows to explore the limitations of the spectral independence assumption and how to generalize it. Overall, these results provide insights on how the ICM principle can be assessed mathematically to infer direction of causation in empirical time series.

**Keywords:** Time Series, Independence of Causal Mechanisms, Information Geometry, Concentration of Measure, Cause-effect Pairs.

## 1. Introduction

One purpose of causal inference is to estimate the direction of cause-effect relationships between different parts of a system, to provide insights about the underlying mechanisms and how to intervene on them to influence the overall behavior. Since interventions are often challenging, or unethical to perform, a number of causal inference techniques have been developed to infer the causal relationships from observational data only (Spirtes, 2010; Pearl, 2000). For this aim, they rely on key assumptions pertaining to the mechanisms generating the observed data. Classical constraint-based search methods, such as the PC algorithm (Spirtes et al., 1993), address this question by assuming successive observations are independent samples from the same unmanipulated density in order to characterize it and infer compatible causal graphical models. However, these

methods cannot infer the causal direction when the graphical model consists only of two variables. Moreover, observed data from complex natural system are often not i.i.d. and time dependent information reflect key aspects of these systems. On the other hand, causal relations between time series are typically explored via Granger-causality type methods (Granger, 1969), which require causal sufficiency. To overcome this limitation, more recent approaches (Mastakouri et al., 2021) (and references therein) employ Markov condition and causal faithfulness to identify characteristic patterns of conditional (in)dependences that witness causal influence even in the presence of hidden common causes.

Most common practical implementations of Granger causality rely on assumptions involving vector autoregressive structural equations and exogenous i.i.d. random variables called the *innovations* of the process (Granger, 1969; Peters et al., 2013). These methods can successfully estimate causal relationships when empirical data is generated according to the assumptions, but the results can be misleading when the model is misspecified. In particular, Granger causality may fail to infer the true direction of causation when the sampling of the time series is not fast enough to capture the dynamical interactions precisely (Geweke, 1982; Gong et al., 2015), an issue that also spoils approaches like (Mastakouri et al., 2021), which also –like Granger– relies on observations of ground truth dynamics at well-defined points in time.

A different approach for inferring causal directions in time series, the Spectral Independence Criterion (SIC), has been introduced in (Shajarisales et al., 2015). In contrast to the above mentioned Granger-like methods, SIC relies on the philosophical principle that the mechanism that generates the observed cause variable and the mechanism that generates the effect from the cause are chosen independently by Nature, such that these two mechanisms do not inform each other. Possible formalizations of this postulate of *Independence of Causal Mechanisms* (ICM), have been proposed in (Janzing and Schölkopf, 2010; Lemeire and Janzing, 2012) (where 'independence' amounts to algorithmic independence) and (Schölkopf et al., 2012) (where 'independence' amounts to semi-supervised learning being useless). These abstract and general ideas have been further exploited to design several concrete domain-specific causal inference methods (Daniusis et al., 2010; Janzing et al., 2012, 2010a; Zscheischler et al., 2011; Sgouritsa et al., 2015; Shajarisales et al., 2015) among which SIC was the first to address the case of time series. It introduces an equation to formulate the ICM postulate when both the cause and the effect are stationary time series and the cause generates the effect trough a linear time invariant mechanism. This framework leads to defining a quantity called Spectral Density Ratio (SDR), quantifying to which extent the ICM postulate is approximately satisfied. The SDR is exploited to decide the direction of causation when a pair of time series is considered, notably in the context of neural times series (Shajarisales et al., 2015; Ramirez-Villegas et al., 2021), as well as to assess the ICM and extrapolation capabilities of convolutional generative models (Besserve et al., 2021). Despite these works, the theoretical underpinnings of SIC remain largely unexplored.

The present paper aims at establishing identifiability results as well as connections to other existing frameworks in causality. After introducing the basic definitions of our causal inference framework (Sec. 2), we introduce the SIC assumption in Sec. 3. We then provide an information geometric perspective on SIC (Sec. 4), as well as theoretical guaranties for identifiability of the causal direction (Sec. 5) that are robust to time-series downsampling (Sec. 6). Finally, an invariance perspective on SIC allows to define generalizations of SIC adapted to specific application domains (Sec. 7). All proofs are provided in Appendix A. Overall, these result clarify the conditions under which SIC is applicable, its limitations and potential extensions.

## 2. Background and Model description

### 2.1. Deterministic cause-effect inference setting

Cause-effect inference based on the ICM principle can be explained as follows. We assume we have two observables $X$ and $Y$, possibly multidimensional and neither necessarily from a vector space, nor necessarily random. We assume additionally that these two observations have a deterministic, acyclic, unconfounded relation: they can be mapped to each other via some invertible map $f$, and their generative model obeys one of the following structural equations:

$$Y := f(X) \text{ or } X := f^{-1}(Y).$$

In the remainder of the paper, we will assume that exactly one of these causal models is the true one (that means we exclude the possibility of a confounder, for example). Without loss of generality, $X \to Y$ is considered to be the true generative model, called the *forward causal model*, while $Y \to X$ will be called the *backward model*. We will call $f$ the mechanism, $X$ the cause and $Y$ the effect. Identifying that the forward model is the true one from observations of $X$ and $Y$ only, is challenging since both models provide a satisfactory description of the data. As a consequence, an additional assumption needs to be added to this setting to inference the direction of causation. The ICM framework relies on the eponymous postulate that properties of $X$ and $f$ are "independent" in some sense. ICM-based methods explore specific settings for which, if the assumption is valid for the true generative model, the converse independence assumption (between $f^{-1}$ and $Y$) is very unlikely. As a consequence, the true causal direction can be identified by evaluating this independence in both scenarios and pick the direction that is closer to satisfying the idealized independence assumption. The remainder of this section will introduce an ICM setting which is specific to time series.

### 2.2. Linear filters

We assume that the ground truth causal mechanism which links a cause time series **x** to an effect time series **y**, is a (deterministic) linear time invariant filter, such that

$$\mathbf{y} = \{y_t\} := \{\sum_{\tau \in \mathbb{Z}} h_{\mathbf{x} \to \mathbf{y}, \tau}\, x_{t-\tau}\} = \mathbf{h}_{\mathbf{x} \to \mathbf{y}} * \mathbf{x}, \tag{1}$$

where **h** denotes the *impulse response* of the filter and $*$ denotes discrete time convolution. Note that this assumption implies that the causal relation is unconfounded. We assume that the filter satisfies the Bounded Input Bounded Output (BIBO) stability property (Proakis, 2001), stating that any bounded input **x** (such that $\sup_t |x_t| < +\infty$) results in a bounded output. In our setting, a necessary and sufficient condition for BIBO stability is $\mathbf{h}_{\mathbf{x} \to \mathbf{y}} \in \ell^1(\mathbb{Z})$, i.e. $\sum_t |h_{\mathbf{x} \to \mathbf{y}, t}| < +\infty$. A filter **h** is called *causal* whenever $h_\tau = 0$ for $\tau < 0$ and *Finite Impulse Response* (FIR) when the transfer function has a finite support.

### 2.3. Stationary sequences

We assume that the input time series **x** is a sample drawn from a real-valued zero mean weakly stationary process (Brockwell and Davis, 2009), $\{X_t, t \in \mathbb{Z}\}$, and denote by $C_{xx}(\tau) = \mathbb{E}[X_t X_{t+\tau}]$ the *autocovariance function* of the process, which does not depend on $t$ due to stationarity. We also assume that $\sum_{\tau \in Z} |C_{xx}(\tau)| < +\infty$ such that its *Power Spectral Density* (PSD) $S_{xx}$, defined as the

Discrete Time Fourier Transform (DTFT) (Vetterli et al., 2014, Chapter 3) of the autocovariance function, is a well defined 1-periodic function of normalized frequency $\nu \in \mathbb{R}$. Background on the DTFT can be found in Appendix B.1. Under these assumptions, $S_{xx}$ is bounded, continuous, and its average over the unit length intervale $\mathcal{I} = [-\frac{1}{2}, \frac{1}{2})$ corresponds to the power of the process $\mathcal{P}(\mathbf{X}) = \mathbb{E}(|X_t|^2) = \langle S_{xx} \rangle$ which is thus well-defined and finite. To simplify notations, we will denote by $\langle . \rangle$ the integral (or average) of a function over the unit length interval $\mathcal{I}$, such that $\mathcal{P}(\mathbf{X}) = \langle S_{xx} \rangle$. When such a sequence is fed into a filter of impulse response $\mathbf{h}_{\mathbf{X} \to \mathbf{Y}}$ as defined above, the stochastic output $\mathbf{Y}$ is weakly stationary with summable autocovariance such that

$$S_{yy}(\nu) = |\widehat{\mathrm{h}}_{\mathbf{X} \to \mathbf{Y}}(\nu)|^2 S_{xx}(\nu), \nu \in \mathcal{I}. \tag{2}$$

where $\widehat{\mathrm{h}}_{\mathbf{X} \to \mathbf{Y}}$ denotes the DTFT of the impulse response (see Appendix B.1 for details), called the *frequency response* of the system. This follows from elementary properties of the Fourier transform and Proposition 3.1.2. in (Brockwell and Davis, 2009). If such a BIBO-stable filtering relationship exists in only one direction (i.e. when the frequency response is not invertible at some frequency), it is natural to assign causality to the stable direction (given the Cause-Effect Inference setting described in section 2.1). If BIBO-stable filters and thus impulse responses can be defined for both directions, we can denote them with $\mathbf{h}_{\mathbf{X} \to \mathbf{Y}}$ and $\mathbf{h}_{\mathbf{Y} \to \mathbf{X}}$, respectively, and their Fourier transforms are continuous and linked by

$$\widehat{\mathrm{h}}_{\mathbf{Y} \to \mathbf{X}} = \frac{1}{\widehat{\mathrm{h}}_{\mathbf{X} \to \mathbf{Y}}}.$$

In such a situation, both the forward and backward filtering models are plausible structural causal models to explain the observed time series. Therefore a more sophisticated criterion is needed for causal inference. We focus on this challenging situation in the present work and will require basic assumptions for the causal models in the remainder of the paper, summarized as follows.

**Assumption 1 (Invertible causal model)** *The cause* $\mathbf{X}$ *is a weakly stationary time series with* $C_{xx} \in \ell^1(\mathbb{Z})$ *and such that* $S_{xx}$ *is strictly positive at all frequencies. The mechanism is an invertible BIBO-stable filter with impulse response* $\mathbf{h}_{\mathbf{X} \to \mathbf{Y}}$ *such that its inverse is also BIBO-stable. The effect is defined as*

$$\mathbf{Y} = \mathbf{h}_{\mathbf{X} \to \mathbf{Y}} * \mathbf{X}.$$

## 3. Spectral Independence Criterion (SIC)

### 3.1. Definition

Given the two stationary processes $\mathbf{X} := \{X_t : t \in \mathbb{Z}\}$ and $\mathbf{Y} := \{Y_t : t \in \mathbb{Z}\}$ such that $\mathbf{X}$ causes $\mathbf{Y}$ through a linear filter, the ICM hypothesis introduced by Shajarisales et al. (2015) can be stated as:

**Postulate 1 (Spectral Independence Criterion (SIC))** *A causal model satisfying Assumption 1 satisfies spectral independence whenever*

$$\langle S_{xx} | \widehat{\mathrm{h}}_{\mathbf{X} \to \mathbf{Y}} |^2 \rangle = \langle S_{xx} \rangle \langle | \widehat{\mathrm{h}}_{\mathbf{X} \to \mathbf{Y}} |^2 \rangle. \tag{3}$$

In practice, this postulate is hypothesized to hold only approximately, and we will explain what is meant by that in Section 5. However, in Secs. 3-4, we will develop theoretical results based on the *perfect* spectral independence postulate, that is, where eq. (3) holds exactly. As eq. (2) indicates, the

filter applies an amplifying factor to the input power at each frequency to provide the output power, and eq. (3) makes a statement on the average amplification achieved across frequencies. Indeed, the left hand side of eq. (3) is the average PSD of the output signal $\{Y_t, t \in Z\}$ over all frequencies, i.e. its total power $\mathcal{P}(\mathbf{Y})$. Hence, SIC states that the output power can be computed by applying the frequency-averaged amplifying factor to the input power. This suggests that the amplification implemented by the mechanism is not informed by the values of the input PSD at each frequency. Note that based on (2) and under Assumption 1, the postulate of eq. (3) can be rephrased using the PSDs of $\mathbf{X}$ and $\mathbf{Y}$ alone, i.e.

$$\langle S_{yy} \rangle = \langle S_{xx} \rangle \langle S_{yy}/S_{xx} \rangle . \tag{4}$$

### 3.2. Measuring spectral dependence

Shajarisales et al. (2015) then introduce the scale invariant quantity $\rho_{\mathbf{X} \to \mathbf{Y}}$ measuring the departure from the SIC assumption, i.e. the dependence between input power spectrum and the frequency response of the filter: the Spectral Dependency Ratio (SDR) from $\mathbf{X}$ to $\mathbf{Y}$ is, under Assumption 1, given by

$$\rho_{\mathbf{X} \to \mathbf{Y}} := \frac{\langle S_{yy} \rangle}{\langle S_{xx} \rangle \langle |\widehat{h}_{\mathbf{X} \to \mathbf{Y}}|^2 \rangle} = \frac{\langle S_{yy} \rangle}{\langle S_{xx} \rangle \langle S_{yy}/S_{xx} \rangle} . \tag{5}$$

Moreover $\rho_{\mathbf{X} \to \mathbf{Y}}$ can be written in terms of total signal powers and energy of the impulse response:

$$\rho_{\mathbf{X} \to \mathbf{Y}} = \frac{\mathcal{P}(\mathbf{Y})}{\mathcal{P}(\mathbf{X}) \| \mathbf{h}_{\mathbf{X} \to \mathbf{Y}} \|_2^2} . \tag{6}$$

To interpret (6), note that the filter $\mathbf{h}_{\mathbf{X} \to \mathbf{Y}}$ amplifies the power of a white noise, i.e. a stationary stochastic process whose power spectrum is constant, by a factor $\| \mathbf{h}_{\mathbf{X} \to \mathbf{Y}} \|_2^2$. Thus, the SDR measures how much the filter amplifies the power of the cause signal compared to the case in which the cause was a white noise. We then define $\rho_{\mathbf{Y} \to \mathbf{X}}$ by exchanging the roles of $\mathbf{X}$ and $\mathbf{Y}$ in the above equations. We provide a probabilistic interpretation of spectral independence in Appendix B.2 and an intuitive example illustrating its meaning in Appendix B.3.

## 4. Information geometric interpretation

While Shajarisales et al. (2015) elaborated on the connection of SIC with the Trace Method (Janzing et al., 2010b), we now investigate its relation to another type of ICM-based approch: Information-Geometric Causal Inference (IGCI) (Daniusis et al., 2010). We get inspiration from Janzing et al. (2012) who established a connection between IGCI for linear relationships and the Trace Method. At the heart of this derivation lies an information geometric interpretation of the principle of independence of cause and mechanism for probability distributions. After introducing this view, we will show how SIC can be casted into the same framework, in the context of Gaussian processes.

### 4.1. Information geometry background

Information Geometry is a discipline where ideas from differential geometry are applied to probability theory. Probability distributions are represented as points from a Riemannian manifold, known

as *statistical manifold*. Equipped with Kullback-Leibler divergence as premetric[1], one can study the geometrical properties of the statistical manifold. For more on this, see e.g. (Amari and Nagaoka, 2007).

For two probability distributions $P, Q$, $D(P\|Q)$ will denote their Kullback-Leibler divergence, also called the relative entropy distance. Given a deterministic causal structural equation of the form $Y := f(X)$, and given $P_X$ and $P_Y$, the distributions of the cause and effect, respectively, we assume the irregularity of each distribution can be quantified by evaluating their divergence to a reference set $\mathcal{E}$ of "regular" distributions[2]

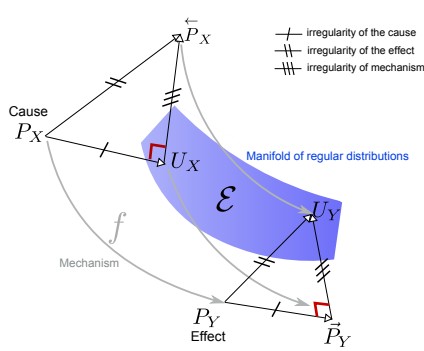

Figure 1: Illustration of eq. (7). Matching symbols on the segments indicate congruency of KL-divergence values.

$$D(P_X\|\mathcal{E}) = \inf_{U\in\mathcal{E}} D(P_X\|U),$$

$$D(P_Y\|\mathcal{E}) = \inf_{U\in\mathcal{E}} D(P_Y\|U).$$

We assume moreover that these infima are reached at a unique point, their projection on $\mathcal{E}$

$$U_X = \arg\min_{U\in\mathcal{E}} D(P_X\|U), \ U_Y = \arg\min_{U\in\mathcal{E}} D(P_Y\|U).$$

Assuming cause $X$ and effect $Y$ are $n$-dimensional multivariate Gaussian random vectors linked by a deterministic linear map $A$, Janzing et al. (2010b) introduced the ICM-based Trace Condition, that we restate in this restricted setting of equal dimension of cause and effect:

**Postulate 2 (Trace Condition)** *Let* tr *denote the matrix trace. The respective covariance matrices of cause and effect,* $\Sigma_X$ *and* $\Sigma_Y$, *satisfy*

$$\mathrm{tr}[\Sigma_Y] = \frac{1}{n}\mathrm{tr}[AA^\top]\mathrm{tr}[\Sigma_X].$$

If we assume $\mathcal{E}$ is the set of isotropic Gaussian distributions, the Trace Condition has been shown to be equivalent to:

$$D(P_Y\|U_Y) = D(P_Y\|\overset{\rightarrow}{P}_Y) + D(\overset{\rightarrow}{P}_Y\|U_Y)$$

where $\overset{\rightarrow}{P}_Y$ is the distribution of $f(X) := AX$ when $X$ is distributed according to $U_X$, where $U_X$ is the isotropic Gaussian distribution with the same variance as $X$. This relation can be interpreted as an orthogonality principle by considering the Kullback-Leibler divergences as a generalization of the squared Euclidean norm for the geodesics $P_Y U_Y$, $P_Y\overset{\rightarrow}{P}_Y$ and $\overset{\rightarrow}{P}_Y U_Y$ as illustrated in Fig. 1. Since applying the bijection $f^{-1}$ preserves the divergences, we get the equivalent relation

$$D(P_Y\|U_Y) = D(P_X\|U_X) + D(\overset{\rightarrow}{P}_Y\|U_Y). \tag{7}$$

---

1. A premetric on a set $\mathcal{X}$ is a function $d : \mathcal{X} \times \mathcal{X} \to \mathbb{R}^+ \cup \{0\}$ such that (i) $d(x,y) \geq 0$ for all $x$ and $y$ in $\mathcal{X}$ and (ii) $d(x,x) = 0$ iff $x = 0$. Unlike a metric, it is not required to be symmetric.
2. Here "regular" is only meant in an intuitive sense, not implying any further mathematical notion. If $\mathcal{E}$ is the set of Gaussians, for instance, the distance from $\mathcal{E}$ measures non-Gaussianity.

This orthogonality principle thus reflects the additivity of irregularities, in the sense that $D(P_Y\|U_Y) = D(P_Y\|\mathcal{E})$ corresponds to the irregularity of $Y$. It measures the distance to the set of "regular" distributions, while $D(P_X\|U_X) = D(P_X\|\mathcal{E})$ measures the irregularity of $X$ in the same way. In addition, it can be shown that $D(\overrightarrow{P}_Y\|U_Y) = D(\overrightarrow{P}_Y\|\mathcal{E})$ and it thus measures the irregularity of the mechanism $f$ indirectly, via the "irregularity" of the distribution resulting from applying $f$ to a regular distribution $U_Y$. We now show that spectral independence encodes a similar relation for discrete time stationary Gaussian processes.

### 4.2. Information geometry of stochastic processes

The generalization of KL-divergence to a pair of stationary time series $(\mathbf{X}, \mathbf{Y})$ is called the relative entropy rate defined as (Ihara, 1993)

$$\bar{D}(P_\mathbf{X}\|P_{\tilde{\mathbf{X}}}) := \lim_{N\to+\infty} \frac{1}{N} D(P_{\mathbf{X}_{1:N}}\|P_{\mathbf{Y}_{1:N}}).$$

where $\mathbf{X}_{i:j}$ is the subvector of vector $\mathbf{X}$ from element $i$ to element $j$. In the case of stationary Gaussian processes, this quantity can be computed explicitly.

**Proposition 1** *Let $\mathbf{X}$ and $\tilde{\mathbf{X}}$ be zero mean weakly stationary discrete Gaussian processes with PSD's $S_{xx}$ and $S_{\tilde{x}\tilde{x}}$, respectively, with $C_{xx}, C_{\tilde{x}\tilde{x}} \in \ell^1(\mathbb{Z})$ and such that $S_{xx}, S_{\tilde{x}\tilde{x}}$ are strictly positive at all frequencies (see also Assumption 1) the relative entropy rate is given by*

$$\bar{D}(P_\mathbf{X}\|P_{\tilde{\mathbf{X}}}) = \frac{1}{2}\int_{-\frac{1}{2}}^{\frac{1}{2}} \left(\frac{S_{xx}(\nu)}{S_{\tilde{x}\tilde{x}}(\nu)} - 1 - \log\frac{S_{xx}(\nu)}{S_{\tilde{x}\tilde{x}}(\nu)}\right)d\nu \,. \tag{8}$$

The proof is a direct application of (Ihara, 1993, Theorem 2.4.4.).

### 4.3. SIC as an orthogonality principle

We consider $\bar{\mathcal{E}}$ the manifold of Gaussian white noises: i.e. Gaussian processes with constant PSD, as the set of regular distributions replacing $\mathcal{E}$. Let $\mathbf{P}_{\bar{\mathcal{E}}}\mathbf{X}$ be the information geometric projection of $\mathbf{X}$ on $\bar{\mathcal{E}}$. This is then a Gaussian white noise of power $\mathcal{P}(\mathbf{X})$ (see Lemma 2). We then have:

**Theorem 1** *Let Assumption 1 hold and further assume that $\mathbf{X}$ is a Gaussian process. Set $U_\mathbf{X} = \mathbf{P}_{\bar{\mathcal{E}}}\mathbf{X}$, $U_\mathbf{Y} = \mathbf{P}_{\bar{\mathcal{E}}}\mathbf{Y}$, and let $\overrightarrow{P}_\mathbf{Y}$ be the distribution obtained by feeding a white noise process with distribution $U_\mathbf{X}$ into $\mathcal{S}$ (thus convolving by $h_{\mathbf{X}\to\mathbf{Y}}$). Then we have:*

$$\bar{D}(P_\mathbf{Y}\|\bar{\mathcal{E}}) = \bar{D}(P_\mathbf{X}\|\bar{\mathcal{E}}) + \bar{D}(\overrightarrow{P}_\mathbf{Y}\|U_\mathbf{Y}) + \frac{1}{2}\left(1 - \frac{\langle S_{xx}\rangle\langle\frac{S_{yy}}{S_{xx}}\rangle}{\langle S_{yy}\rangle}\right). \tag{9}$$

As a consequence the following corollary shows that SIC corresponds to orthogonality of PSD variations in information space, in the case of weakly stationary Gaussian processes.

**Corollary 1** *Under assumptions of Theorem 1, the SIC postulate is equivalent to the orthogonality of irregularities relative to white-noise Gaussian processes:*

$$\bar{D}(P_\mathbf{Y}\|\bar{\mathcal{E}}) = \bar{D}(P_\mathbf{X}\|\bar{\mathcal{E}}) + \bar{D}(\overrightarrow{P}_\mathbf{Y}\|\bar{\mathcal{E}}). \tag{10}$$

## 5. Identifiability results

The proposal by Shajarisales et al. (2015) to use SIC for inferring the causal direction is essentially based on two insights: (i) an argument for why the SDR is expected to be close to 1 for the *causal* direction; (ii) an argument for why the SDR is *not* expected to be close to 1 in the anti-causal direction. After reviewing these arguments, we show for the first time that they can be exploited to show identifiability of the causal direction based on SDR in the following toy generative model.

### 5.1. Generative model

Assume a length $m$ Finite Impulse Response (FIR) $\mathbf{h}$, such that $h_\tau = 0$ for all $\tau < k$ and all $\tau \geq k + m$, for some $m$ and $k$. Then $\mathbf{h}$ is parametrized by an $m$-dimensional real vector $\mathbf{b}$ such that $h_{i+k} = b_i \quad i = 0, \ldots, m - 1$. We assume that $\mathbf{b}$ has been generated by nature as a single sample drawn from a random variable $\mathbf{B}$ with a spherically symmetric distribution (see for example (Bryc, 2012, Chapter 4) for a rigorous definition). This implies $\mathbf{B}$ takes the form

$$\mathbf{B} = R\mathbf{U},$$

where $R \geq 0$ is a real valued radius random variable, and $\mathbf{U}$ is a random vector uniformly distributed on the unit sphere in $\mathbb{R}^m$. $\mathbf{U}$ and $R$ are statistically independent (Bryc, 2012, Theorem 4.1.2), which entails that for any orthogonal transformation $T \in O(m)$, the distribution of $T\mathbf{b}$ is identical to the distribution of $\mathbf{b}$. An important family of spherically symmetric distributions is the zero mean isotropic gaussians (i.e. with a covariance matrix that is a multiple of the identity matrix). The general idea behind this assumption is that the distribution of the impulse response is invariant to a reparameterization of the vector space in a new system of coordinates that preserves the energy of the impulse response. Theorem 2 shows that for large $m$ the resulting filter will approximately satisfy SIC with high probability.

**Theorem 2 (Concentration of measure for FIR filters)** *Assume a length $m$ mechanism's impulse response $\mathrm{h}_{\mathbf{X} \to \mathbf{Y}}$ with a spherically symmetric generative model and a fixed input PSD $S_{xx}$, then for any given $\varepsilon > 0$, with probability at least $1 - 2\exp(-m\varepsilon^2)$, we have*

$$|\rho_{X \to Y} - 1| \leq \frac{8\varepsilon}{\langle S_{xx} \rangle} \max_\nu S_{xx}(\nu).$$

The exponential term of the probability bound shows that if $m$ is large enough, we can get a tight bound around 1 for the forward SDR with high probability. This kind of *concentration of measure* result has been provided by Shajarisales et al. (2015) (Theorem 1 therein) based on Levy's Lemma. In comparison, the above Theorem 2 using a different random matrix theory result (Guionnet, 2009, Corollary 6.14) resulting notably in a different multiplicative constant in the exponential.

### 5.2. Forward-backward inequality

The following result from (Shajarisales et al., 2015) shows additionally that the dependence measures in both directions are related, as their product can be bounded as a function of the coefficient of variation of $|\widehat{\mathrm{h}}_{\mathbf{X} \to \mathbf{Y}}|^2$ along the frequency axis, defined as the ratio of the standard deviation to the mean

$$CV(|\widehat{\mathrm{h}}_{\mathbf{X} \to \mathbf{Y}}|^2) := \frac{\left\langle \left(|\widehat{\mathrm{h}}_{\mathbf{X} \to \mathbf{Y}}|^2 - \langle |\widehat{\mathrm{h}}_{\mathbf{X} \to \mathbf{Y}}|^2 \rangle \right)^2 \right\rangle^{1/2}}{\langle |\widehat{\mathrm{h}}_{\mathbf{X} \to \mathbf{Y}}|^2 \rangle}.$$

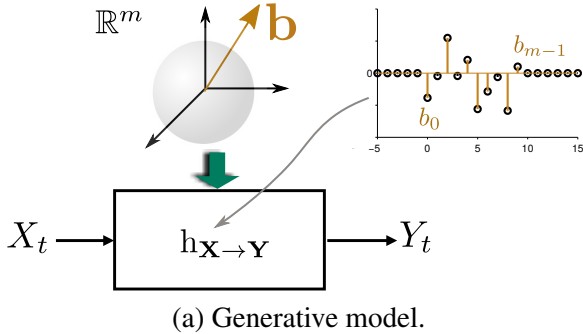

(a) Generative model.

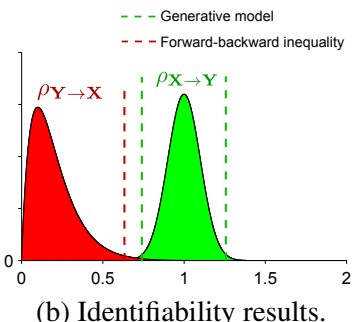

(b) Identifiability results.

Figure 2: (a) Principle of the generative model. (b) Illustration of the two steps leading to identifiability: the forward SDR concentrates around 1, and the forward-backward inequality bounds the backward SDR away from one. Dashed lines indicate high-probability bounds for the SDR values.

**Proposition 2 (Forward-backward inequality)** *For a given linear filter $\mathcal{S}$ with impulse response $\mathbf{h_{X\to Y}}$, input PSD $S_{xx}$ and output PSD $S_{yy}$, let $\mathcal{S}$ have a non-constant modulus frequency response $\widehat{\mathbf{h}}_{X\to Y}$, and assume there exists $\alpha > 0$ such that for all $\nu \in \mathcal{I}$,*

$$|\widehat{\mathbf{h}}_{\mathbf{X\to Y}}(\nu)|^2 \leq (2-\alpha)\left\langle|\widehat{\mathbf{h}}_{\mathbf{X\to Y}}|^2\right\rangle.$$

*Then*

$$\rho_{\mathbf{X\to Y}}\cdot\rho_{\mathbf{Y\to X}} \leq \left[1+\alpha CV\left(|\widehat{\mathbf{h}}_{\mathbf{X\to Y}}|^2\right)^2\right]^{-1} < 1. \tag{11}$$

Note that $\alpha < 1$ because a random variable cannot be everywhere smaller than its mean. According to (11), large fluctuations of $|\widehat{\mathbf{h}}_{\mathbf{X\to Y}}|^2$ around its mean guarantee that the product of the independence measures will be bounded away from 1. Assuming the SIC postulate is satisfied in the forward direction such that $\rho_{\mathbf{X\to Y}} = 1$, it follows naturally that $\rho_{\mathbf{Y\to X}} < 1$.

### 5.3. Inference algorithm

Based on the above two steps, Shajarisales et al. (2015) have introduced a SIC-based algorithm to infer the direction of causation. First, Theorem 2 guaranties that $\rho_{\mathbf{X\to Y}}$ concentrates around one, such that with high probability $1 - \xi \leq \rho_{\mathbf{X\to Y}} \leq 1 + \xi$ (the closer is $\xi$ to zero, the tighter is the bound) with $\xi > 0$. Second, Proposition 2 implies a lower bound for the backward SDR based on the foward SDR value. Indeed, using Proposition 2, for any $\eta > 0$, $\rho_{\mathbf{Y\to X}}\rho_{\mathbf{X\to Y}} < 1 - \eta$ leads to

$$\rho_{\mathbf{Y\to X}} < \frac{1-\eta}{\rho_{\mathbf{X\to Y}}^2}\rho_{\mathbf{X\to Y}}.$$

Thus, combining both results, we get $\rho_{\mathbf{Y\to X}} < \frac{1-\eta}{(1-\xi)^2}\rho_{\mathbf{X\to Y}}$ and obtain that $\rho_{\mathbf{Y\to X}}$ is strictly smaller than $\rho_{\mathbf{X\to Y}}$ with if we can have $1 - \eta < (1-\xi)^2$. This qualitative reasoning, illustrated in Fig. 2(b), suggest that a good strategy to infer the direction of causation is to choose the direction with largest SDR. The corresponding causal inference algorithm is described in Algorithm 1.

### 5.4. Identifiability of the generative model

Proposition 2 argues that SDR is bounded away from 1 for the backward direction, given that the variation of $h$ satisfies some conditions. This way, we guarantee identifiability of causal directions. Since we have not explored whether the generating model of Section 5.1 guarantees these conditions to be true, we now show identifiability for this model separately. While the previous

---
**Algorithm 1** SIC_Inference
---
1: **procedure** SIC_INFERENCE($\mathbf{X},\mathbf{Y}$)
2:     Calculate $\rho_{\mathbf{X}\to\mathbf{Y}}$ and $\rho_{\mathbf{Y}\to\mathbf{X}}$ using (5)
3:     **if** $\rho_{\mathbf{X}\to\mathbf{Y}} > \rho_{\mathbf{Y}\to\mathbf{X}}$ return $\mathbf{X}\to\mathbf{Y}$
4:     **else** return $\mathbf{Y}\to\mathbf{X}$
---

justification provides insights about how Algorithm 1 can identify the direction of causation by bounding forward and backward SDR's, they do not guaranty identifiability of the above generative model using SIC-based causal inference, which turns out to be non-trivial. Identifiability can be proved by bounding the gap of the forward-backward inequality (11) as follows.

**Lemma 1** *Suppose the model from Theorem 2 with fixed input PSD $S_{xx}$ is given, and assume that the filter coefficients $b_1, b_2, \ldots, b_m$ are independently drawn from some distribution $P_B$ with $\mathbb{E}[B] = 0$ and variance $\mathbb{E}[B^2] = 1$ and finite $\mathbb{E}[|B|^3]$. Then $\rho_{\mathbf{X}\to\mathbf{Y}}\rho_{\mathbf{Y}\to\mathbf{X}}$ converges to zero in probability for $m \to \infty$.*

This lemma allows to show, for the first time, identifiability with high probability when the number of filter coefficients gets large.

**Theorem 3** *Given the sampling of coefficients in Lemma 1 and the assumptions there, then the probability for $\rho_{\mathbf{X}\to\mathbf{Y}} > \rho_{\mathbf{Y}\to\mathbf{X}}$ tends to 1 for $m \to \infty$ (as the dimension $m$ of the filter increases), i.e. the direction of causation is identifiable with probability converging to 1.*

## 6. Robustness to downsampling

In the context of time series, the issue of robustness of causal discovery methods to resampling has been raised in several papers (Gong et al., 2015; Palm and Nijman, 1984; Harvey, 1990). In particular, Granger causality is known to have issues with downsampling, and specific correction procedures are required (Gong et al., 2015) to infer causal relations. Here we provide a theoretical result supporting that SIC causal inference is robust to the classical decimation procedure performed in discrete signal processing using an ideal anti-aliasing low pass filter (Crochiere and Rabiner, 1981).

### 6.1. Decimation procedure

Our decimation setting is provided in Fig. 3, showing that both the cause and the effect time series are low-pass filtered with an ideal anti-aliasing filter and then decimated by a factor $D$. The frequency response of the ideal anti-aliasing filter satisfies

$$|\hat{\mathbf{a}}(\nu)| = \mathbb{1}_{|\nu|<1/(2D)}, \; |\nu| \leq 1/2,$$

and the decimation blocks convert a signal $\mathbf{s} = \{s_k\}_{k\in\mathbb{Z}}$ into $\tilde{\mathbf{s}} = \downarrow_D \mathbf{s} = \{s_{kD}\}_{k\in\mathbb{Z}}$ (i.e. by picking the value of one time point of every D points). Using classical decimation results (Crochiere and Rabiner, 1981), we can derive that

$$S_{\tilde{x}\tilde{x}}(\nu) = \frac{1}{D}\sum_{k=-\infty}^{\infty} S_{a*x\,a*x}\left(\frac{\nu}{D} - \frac{k}{D}\right) = \frac{1}{D}\sum_{k=-\infty}^{\infty} S_{xx}\left(\frac{\nu}{D} - \frac{k}{D}\right)\left|\hat{\mathbf{a}}\left(\frac{\nu}{D} - \frac{k}{D}\right)\right|^2$$

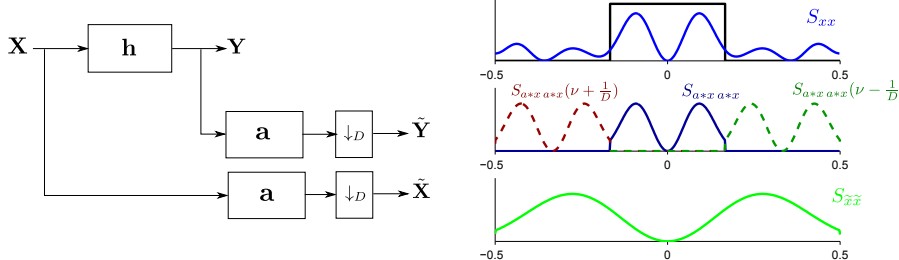

Figure 3: Left: Schema of the decimation procedure. Right: Illustration of how the decimation and ideal anti-aliasing filter affects the input PSD.

Noticing the absence of overlap (see illustration Fig. 3 between the support of each term of the above periodic summation), we get

$$S_{\widetilde{x}\widetilde{x}}(\nu) = \frac{1}{D}S_{xx}\left(\frac{\nu}{D}\right) \text{ and } S_{\widetilde{y}\widetilde{y}}(\nu) = \frac{1}{D}S_{yy}\left(\frac{\nu}{D}\right), \nu \in [-1/2,\, 1/2].$$

We will now investigate whether the true causal direction $\mathbf{X} \to \mathbf{Y}$ can be inferred from the decimated observations only.

## 6.2. Identifiability of the decimated model

Although the decimation certainly involves a loss of information of the measured time series, the following theorem suggests that the SIC can be preserved by such an operation.

**Theorem 4** *Assume the forward generative model of the previous section, and $(\widetilde{\mathbf{X}}, \widetilde{\mathbf{Y}})$ the time series resulting from the decimation of this model by an integer factor $D$ using an ideal anti-aliasing filter. Then for any given $\epsilon$ with $0 < \epsilon < \frac{1}{4D}$ we have*

$$|\rho_{\widetilde{\mathbf{X}}\to\widetilde{\mathbf{Y}}} - 1| \le \epsilon\left(K + (1 + \epsilon K)\frac{2}{1 - 4D\epsilon}\right),$$

*with probability $\delta := (1 - \exp(-\kappa(m-1)\epsilon^2))^2$, where $\kappa$ is a positive global constant (independent of $m$, and $\epsilon$) and $K = \frac{\max_{|\nu|<1/2D} S_{xx}(\nu)}{\int_0^{1/2D} S_{xx}(\nu)d\nu}$.*

As a consequence of Theorem 4, even after decimating the signal, the concentration of the forward SDR around one is guarantied for high dimensional impulse responses. Note also that the forward-backward inequality also holds for decimated data, such that the overall identifiability properties are preserved. However, as intuitively expected, identifiability for an increasing decimation factor $D$ will progressively deteriorate, as the decimation procedure stretches the input and output PSDs from the interval $[0,\, 1/2]$ to $[0,\, 1/(2D)]$. The estimated frequency response is then also stretched, and its coefficient of variation $CV(|\hat{\mathbf{h}}_{\widetilde{\mathbf{X}}\to\widetilde{\mathbf{Y}}}|^2)$ converges to zero as $D$ increases, provided the frequency response respects some smoothness assumptions (e.g. bounded derivative), deteriorating the bound of the forward-backward inequality and thus making the identification of the true causal direction harder. Qualitatively, indentifiability properties are preserved as long as $S_{xx}$ has enough variance on the interval $[0,\, 1/(2D)]$.

## 7. Extension of SIC through invariance principles

While a coherent theoretical framework has been presented in the above sections, whether spectral independence is a valid assumption in a given empirical setting remains to be addressed. Notably, one can question the choice of white noise regular distributions introduced in Section 4. Indeed, the departure from this reference is exploited to quantify orthogonality, and as such, implicitly reflect an assumption about the considered problem. To shed light on this assumption, the group invariance perspective on ICM is helpful (Besserve et al., 2018), and we develop it for the case of SIC in Appendix C. This view justifies the concept of whitening as a way to adapt the SIC framework to datasets where a different set of regular distributions is more appropriate. To ease readability, we justify this whitening in main text based on the following informal arguments. The IGCI perspective of Sec. 4 shows that, by application of corollary 1, causes/inputs that belong to the set of regular distributions trivially satisfy SIC for any choice of mechanism (i.e. filter). These regular distributions satisfy an invariance property: PSDs are invariant to translations along the frequency axis. It is thus natural to consider that SIC is suited to applications where signals with such invariance are uninformative. In contrast, this is not suited for the cases in which such invariance is atypical, as the irregularities quantified by SIC will be blind to this information.

### 7.1. SIC for power law biological signals

Many biological signals exhibit power law-like PSDs, which decay proportionally to a positive power of the frequency. This is particularly the case for brain electrical activity (He et al., 2010). Translation invariant PSDs appear clearly as atypical in such contexts. However, this can be fixed by preprocessing the recorded signals using an invertible *whitening filter* $\mathbf{w}$ such that its squared frequency response $|\widehat{\mathbf{w}}(\nu)|^2$ applies an amplifying factor to each frequency that corrects the tendency of the observed signal to have lower power in high frequencies. This procedure generates a proper group invariance framework as introduced by Besserve et al. (2018), and corresponds to replacing white noises in Sec. 4.3 by regular distributions with PSDs of the form

$$S_{\mathbf{w}} = \frac{\gamma}{|\widehat{\mathbf{w}}(\nu)|^2},\ \gamma > 0\,,$$

which now are representative/typical of the power-law behavior of the considered signals. SIC can then simply be applied to the preprocessed signal in order to assess ICM in a way that is adapted to electrical brain activity.

### 7.2. Experiments

To test this approach, we use the neural recording dataset studied in (Shajarisales et al., 2015). In short, these are recordings of Local Field Potentials (LFP) from two subfields of the rat hippocampus, CA3 and CA1, known to have a clear directed monosynaptic connections CA3$\rightarrow$ CA1 that we thus consider as the ground truth causal relationship. Figure 4(a) depicts the PSD values for recordings from several electrodes in CA3 computed using Welch's method. The empirical distribution of PSD values form this sample do not follow a (frequency)-translation invariant distribution, as the higher frequencies have a much lower power (note the logarithmic scale of the plot). To correct this property, we apply the above-described whitening transformation, by choosing the inverse of the empirical average PSD over the full dataset (including both recordings form CA3 and CA1.

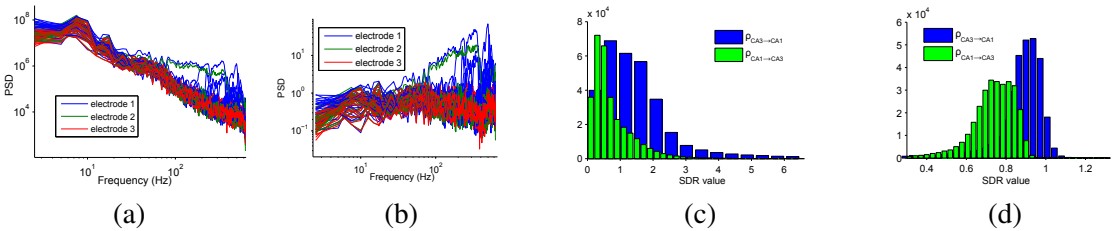

Figure 4: SIC analysis of hippocampal neural data. (a) PSD before whitening. (b) PSD after whitening. (c) SDR before whitening. (d) SDR after whitening.

Examples of the resulting whitened PSDs are shown on Fig. 4(b), where we can observe PSD imbalance across frequencies is largely attenuated, suggesting the distribution of whitened PSDs is closer to being frequency translation invariant. To apply the SIC causal inference approach, we estimated the SDR in both causal (CA3→CA1) and anti-causal (CA1→CA3) directions. While the SDR distribution of original data in the causal direction is widespread (Fig. 4(c)), it becomes much more concentrated around 1 after whitening (Fig. 4(d)), suggesting that the SIC assumption is "on average valid" after whitening the signals, in line with eq. (26) in Appendix C. The SDR distribution in the anticausal direction also becomes bounded to values strictly inferior to 1, as predicted by identifiability results (Shajarisales et al., 2015). Finally, the performance of causal inference using SIC before and after whitening is compared. The distribution of average performance results across time for all pairs of electrodes is represented by box plots on Supplemental Fig. 8, showing that after whitening the performance is more consistent across electrodes than before: lower tale of the distribution spreads much less towards the low performance values, and average increases significantly from 69.0% to 82.8% after whitening (significant paired signed rank test, $p < 10^{-25}$).

## 8. Discussion

We investigated theoretical foundations of the SIC postulate. The information geometric view shows that SIC formalizes an independence of input and mechanism via an 'orthogonality of irregularities' relative to a set of regular distributions of Gaussian white noises. We further show that SIC allows identifying the true causal direction in a toy setting, and that this criterion is robust to downsampling. Finally, we show that the choice of 'regular distribution' can be adapted to the application domain. This set of results clarifies the conditions under which SIC appropriately infers causality based on empirical data. Extending the framework to systems with noise, non-linearities, hidden confounders and multiple dimensions are key next steps to establish this methodology as a standard tool for time series analysis.

## Acknowledgments

This work was partially supported by the German Federal Ministry of Education and Research (BMBF): Tübingen AI Center, FKZ: 01IS18039B; and by the Machine Learning Cluster of Excellence, EXC number 2064/1 - Project number 390727645.

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

## Appendix A. Proofs

This section includes proofs of main text results and additional necessary results.

### A.1. Consequence of Assumption 1

Due to basic properties of the Fourier transform, we have the following implications for the model in Assumption 1.

**Proposition 3** *Assumption 1 implies*

- *The input and output PSDs $S_{xx}$ and $S_{yy}$ are real 1-periodic bounded continuous functions taking values on $[a, \max_{\nu \in \mathcal{I}} S_{xx}(\nu)]$ and $[b, \max_{\nu \in \mathcal{I}} S_{xx}(\nu)]$ respectively, for some $a, b > 0$.*

- *The forward and backward frequency responses $\widehat{h}_{\mathbf{X} \to \mathbf{Y}}$ and $\widehat{h}_{\mathbf{Y} \to \mathbf{X}}$ are complex bounded 1-periodic continuous functions such that $|\widehat{h}_{\mathbf{X} \to \mathbf{Y}}|^2 = \frac{S_{yy}}{S_{xx}} > c$ and $|\widehat{h}_{\mathbf{X} \to \mathbf{Y}}|^2 = \frac{S_{xx}}{S_{yy}} > d$ for some $c, d > 0$.*

**Proof** For any sequence $\mathbf{a} \in \ell^1(\mathbb{Z})$ there is uniform convergence and boundedness of the Fourier transform series (Vetterli et al., 2014, Chapter 3). As a consequence, all Fourier transforms considered are well defined, periodic, continuous and bounded. $S_{xx}$ being strictly positive, by continuity on one period (as Fourier transform of the autocorrelation function), its minimum is strictly positive. BIBO stability of $h_{X \to Y}$ entails that $S_{yy}$ is also well defined, periodic, continuous and bounded (e.g. using the above Proposition 1). Then BIBO stability of $h_{Y \to X}$ entails that its Fourier transform $S_{xx}/S_{yy}$ is bounded, which implies by contradiction that $\inf_v S_{yy}(v)$ must be strictly positive. Finally, strictly positive bounds on $S_{xx}$ and $S_{yy}$ imply strictly positive upper and lower bounds for the frequency responses. ∎

### A.2. Proof of Theorem 1

First, we start by finding the divergence rate for a given Gaussian process from the set of white Gaussian noises (i.e. the Gaussian processes with constant spectra).

**Lemma 2** *Let $\bar{\mathcal{E}}$ to be the set of discrete white Gaussian noises. Assuming $\mathbf{X}$ is a Gaussian process with $C_{xx} \in \ell^1(\mathbb{Z})$ and such that $S_{xx}$ is strictly positive at all frequencies, then the projection $\mathbf{P}_{\bar{\mathcal{E}}}\mathbf{X}$ of $\mathbf{X}$ on $\bar{\mathcal{E}}$ is a Gaussian white noise of power $\mathcal{P}(\mathbf{X})$, and the corresponding divergence is*

$$\bar{D}(P_{\mathbf{X}} \| \bar{\mathcal{E}}) = -\frac{1}{2} \int_{-\frac{1}{2}}^{\frac{1}{2}} \ln \left( \frac{S_{xx}(\nu)}{\mathcal{P}(\mathbf{X})} \right) d\nu \tag{12}$$

*The formula also holds when exchanging $\mathbf{X}$ for $\mathbf{Y}$*

To get a intuition of (12) note that $q(\nu) := \frac{S_{xx}(\nu)}{\mathcal{P}(\mathbf{X})}$ can be formally interpreted as probability density on $[-1/2, 1/2]$ due to $\int_{-1/2}^{1/2} q(\nu) d\nu = 1$. If $u$ with $u(\nu) = 1$ denotes the constant density, (12) can be rephrased as the relative entropy distance $\int_{-1/2}^{1/2} u(\nu) \log(q\nu) u(\nu) = D(u\|q)$. In other words, it measures to what extent $q$ deviates from the uniform distribution.

Based on this Lemma, we follow the steps of (Janzing et al., 2012) to prove the theorem.

**Proof** We denote elements of the set of white Gaussian noise distributions $U_\alpha$, where $\alpha$ denotes the power of $U_\alpha$ and hence the constant value of its corresponding PSD. The Gaussian noise distribution with minimum distance corresponds to the value of $\alpha$ for which the derivative of $\bar{D}(P_{\mathbf{X}}\|U_\alpha)$ vanishes. We can compute the distances via Proposition 1 because condition $(ii)$ is satisfied:

$$\frac{d\bar{D}(P_{\mathbf{X}}\|U_\alpha)}{d\alpha} = \frac{1}{2}\int_{-\frac{1}{2}}^{\frac{1}{2}}\left(-\frac{S_{xx}(\nu)}{\alpha^2} + \frac{S_{xx}(\nu)}{\alpha^2}\frac{\alpha}{S_{xx}(\nu)}\right)d\nu.$$

Hence, the derivative vanishes for $\int_{-\frac{1}{2}}^{\frac{1}{2}}\frac{S_{xx}(\nu)}{\alpha^2} = \frac{1}{\alpha}$, which amounts to $\alpha = \mathcal{P}(\mathbf{X})$. We thus get

$$\bar{D}(P_{\mathbf{X}}\|\mathcal{E}_X) = \frac{1}{2}\int_{-\frac{1}{2}}^{\frac{1}{2}}\frac{S_{xx}(\nu)}{\mathcal{P}(\mathbf{X})} - 1 - \ln\left(\frac{S_{xx}(\nu)}{\mathcal{P}(\mathbf{X})}\right)d\nu.$$

Using the definition of $\mathcal{P}$ we obtain:

$$\frac{1}{2}\int_{-\frac{1}{2}}^{\frac{1}{2}}\left(\frac{S_{xx}(\nu)}{\mathcal{P}(\mathbf{X})} - 1 - \ln\left(\frac{S_{xx}(\nu)}{\mathcal{P}(\mathbf{X})}\right)\right)d\nu = \frac{1}{2}\frac{\int_{-\frac{1}{2}}^{\frac{1}{2}}S_{xx}(\nu)}{\mathcal{P}(\mathbf{X})} - \frac{1}{2} - \frac{1}{2}\int_{-\frac{1}{2}}^{\frac{1}{2}}\ln\left(\frac{S_{xx}(\nu)}{\mathcal{P}(\mathbf{X})}\right)d\nu$$

$$= -\frac{1}{2}\int_{-\frac{1}{2}}^{\frac{1}{2}}\ln\left(\frac{S_{xx}(\nu)}{\mathcal{P}(\mathbf{X})}\right)d\nu,$$

which proves our claim. The formula also holds for $\mathbf{Y}$ because a linear filter maps a Gaussian process on a Gaussian process. ∎

**Proof of Theorem 1:** Using Lemma 2 we have

$$\bar{D}(P_{\mathbf{X}}\|\bar{\mathcal{E}}) = -\frac{1}{2}\int_{-\frac{1}{2}}^{\frac{1}{2}}\ln\left(\frac{S_{xx}(\nu)}{\mathcal{P}(\mathbf{X})}\right)d\nu \qquad \text{and} \qquad \bar{D}(P_{\mathbf{Y}}\|\bar{\mathcal{E}}) = -\frac{1}{2}\int_{-\frac{1}{2}}^{\frac{1}{2}}\ln\left(\frac{S_{yy}(\nu)}{\mathcal{P}(\mathbf{Y})}\right)d\nu.$$

Transforming $U_{\mathbf{X}}$ and $h_t$ to Fourier domain it is easy to see that $\overrightarrow{P_{\mathbf{Y}}}$ is the distribution of a stationary Gaussian process with PSD $\mathcal{P}(\mathbf{X})|\hat{h}(\nu)|^2$ according to eq. (2). Therefore using Proposition 1 we get:

$$\bar{D}(\overrightarrow{P_{\mathbf{Y}}}\|U_{\mathbf{Y}}) = \frac{1}{2}\int_{-\frac{1}{2}}^{\frac{1}{2}}\left(\frac{\mathcal{P}(\mathbf{X})|\hat{h}(\nu)|^2}{\mathcal{P}(\mathbf{Y})} - 1 - \log\frac{\mathcal{P}(\mathbf{X})|\hat{h}(\nu)|^2}{\mathcal{P}(\mathbf{Y})}\right)d\nu,$$

which completes the proof. □

### A.3. Proof of Corrolary 1

**Proof** Assume SIC is satisfied, then the last term of eq. (9) vanishes, yielding

$$\bar{D}(P_{\mathbf{Y}}\|\bar{\mathcal{E}}) = \bar{D}(P_{\mathbf{X}}\|\bar{\mathcal{E}}) + \bar{D}(\overrightarrow{P_{\mathbf{Y}}}\|U_{\mathbf{Y}}).$$

Moreover, SIC also implies that the projection of $\overrightarrow{P_{\mathbf{Y}}}$ on $\bar{\mathcal{E}}$ is $U_{\mathbf{Y}}$. This is because SIC implies that $\overrightarrow{P_{\mathbf{Y}}}$ and $P_{\mathbf{Y}}$ have the same power. Hence, we get

$$\bar{D}(P_{\mathbf{Y}}\|\bar{\mathcal{E}}) = \bar{D}(P_{\mathbf{X}}\|\bar{\mathcal{E}}) + \bar{D}(\overrightarrow{P_{\mathbf{Y}}}\|\bar{\mathcal{E}}).$$

For the converse implication, assume we have orthogonality of irregularities, combining eq. (10) with eq. (9) we get

$$\bar{D}(\vec{P_{\mathbf{Y}}}||\bar{\mathcal{E}}) = \bar{D}(\vec{P_{\mathbf{Y}}}||U_{\mathbf{Y}}) + \frac{1}{2}\left(1 - \frac{\langle S_{xx}\rangle\langle\frac{S_{yy}}{S_{xx}}\rangle}{\langle S_{yy}\rangle}\right).$$

Using Lemma 2 to get an analytic expression for the left-hand side of the above equation, and eq. (8) for the right-hand side this yields

$$-\frac{1}{2}\int_{-1/2}^{1/2}\log\frac{|\hat{h}(\nu)|^2}{\langle|\hat{h}|^2\rangle}d\nu = \frac{1}{2}\int_{-\frac{1}{2}}^{\frac{1}{2}}\left(\frac{\mathcal{P}(\mathbf{X})|\hat{h}(\nu)|^2}{\mathcal{P}(\mathbf{Y})}-1-\log\frac{\mathcal{P}(\mathbf{X})|\hat{h}(\nu)|^2}{\mathcal{P}(\mathbf{Y})}\right)d\nu+\frac{1}{2}\left(1-\frac{\langle S_{xx}\rangle\langle\frac{S_{yy}}{S_{xx}}\rangle}{\langle S_{yy}\rangle}\right)$$

which after simplification, and noting that $\langle|\hat{h}|^2\rangle = \langle\frac{S_{yy}}{S_{xx}}\rangle$, yields

$$\log\langle\frac{S_{yy}}{S_{xx}}\rangle = \log\frac{\mathcal{P}(\mathbf{Y})}{\mathcal{P}(\mathbf{X})},$$

which is equivalent to SIC. ∎

### A.4. Proof of Lemma 1

**Proof** We denote $\hat{h}_b^m(\nu)$ the transfer function $\widehat{h}_{\mathbf{X}\to\mathbf{Y}}$ for the model such that

$$\hat{h}_b^m(\nu) := \frac{1}{\sqrt{m}}\sum_{j=1}^{m}b_j e^{-i2\pi\nu j}.$$

Let $B_1, B_2, ...$ be a sequence of i.i.d. real-valued variables on a probability space $(\Omega, \Sigma, P_\Omega)$. Then we first define for each frequency $\nu$ the sequence $\left(\widehat{B}_\nu^m\right)_{m\in\mathbb{N}}$ of random variables via

$$\widehat{B}_\nu^m := \frac{1}{\sqrt{m}}\sum_{j=1}^{m}B_j(\omega)e^{-i2\pi\nu j}.$$

Using know vector valued central limit theorems, $\widehat{B}_\nu^m$ converges to Gaussian on the complex plane for each $\nu$.

Now define for each $\omega \in \Omega$ the sequence $\left(\widehat{B}^m(\omega)\right)_{m\in\mathbb{N}}$ of random variables on the probability space $(\mathcal{I}, \mathcal{B}, \lambda)$ with $\mathcal{B}$ denoting the Borel sigma algebra and $\lambda$ the Lebesgue measure (to formalize the random choice of frequency), via

$$\widehat{B}^m(\omega) : \nu \to \widehat{B}^m(\omega) = \frac{1}{\sqrt{m}}\sum_{j=1}^{m}B_j e^{-i2\pi\nu j}.$$

Theorem 1 in (Janzing et al., 2017) states that this sequence of complex random variables converges for $m \to \infty$ to an isotropic two-dimensional Gaussian $Z$ on the complex plane where real and imaginary parts are independent with variance $1/2$. More precisely, the random variable

$$\omega \to d(P_{\widehat{B}^m(\omega)}, P_Z)$$

converges in probability to zero for any pseudo-metric $d$ that is well-behaved in the sense of Definition 1 in (Janzing et al., 2017).

We now define a pseudo-metric on the set of distributions on $\mathbb{C}$ by

$$d(P, Q) := \sup_{x \in \mathbb{R}^+} \left| P(|Z|^2 < x) - Q(|Z|^2 < x) \right|. \tag{13}$$

This pseudo-metric is well-defined because

$$d'(P, Q) := \sup_{x \in \mathbb{R}} |P(X < x) - Q(X < x)|$$

is well-behaved for the set of distributions on $\mathbb{R}$, see remarks after Definition 1 in (Janzing et al., 2017).

We now show that

$$\mathbb{E}\left[|\widehat{B}^m(\omega)|^2\right] \mathbb{E}\left[\frac{1}{|\widehat{B}^m(\omega)|^2}\right] = \int_{\mathcal{I}} |\hat{h}_b^m(\nu)|^2 d\nu \cdot \int_{\mathcal{I}} 1/|\hat{h}_b^m(\nu)|^2 d\nu \underset{m \to +\infty}{\to} +\infty$$

in probability. First, note that

$$\int |\hat{h}_b^m(\nu)|^2 d\nu = \frac{1}{m} \sum_{j=1}^{m} b_j^2 \underset{m \to +\infty}{\overset{a.s.}{\to}} \mathbb{E}[B] = 1, \tag{14}$$

due to the strong law of large numbers. Recall that $|Z|^2$ is $\chi^2$-distributed with one degree of freedom, which implies that its density has infinite slope at $0$. Hence, for any arbitrarily large $c$ we can always find a sufficiently small $\epsilon$ such that

$$c\epsilon < P(|Z|^2 \leq \epsilon).$$

Form Markov's inequality for $a > 0$ we get

$$P\left(\frac{1}{|\widehat{B}^m(\omega)|^2} \geq a\right) \leq \frac{1}{a} \mathbb{E}\left[\frac{1}{|\widehat{B}^m(\omega)|^2}\right], \tag{15}$$

such that

$$P\left(|\widehat{B}^m(\omega)|^2 \leq \epsilon\right) \leq \epsilon \mathbb{E}\left[\frac{1}{|\widehat{B}^m(\omega)|^2}\right]. \tag{16}$$

For sufficiently large $m$ we thus can ensure that

$$c\epsilon \leq P\left(|\widehat{B}^m(\omega)|^2 \leq \epsilon\right), \tag{17}$$

with probability at least $1 - \delta$ for any arbitrarily small $\delta$. This is because, as stated above, Theorem 1 in (Janzing et al., 2017) entails that $P_{\widehat{B}^m(\omega)}$ converges to the distribution of $P_Z$ with respect to any well-behaved pseudo-metric, in particular with respect to $d$ defined in (13). Combining (17) and (16) shows that

$$c \leq \mathbb{E}\left[\frac{1}{|\widehat{B}^m(\omega)|^2}\right] = \int_{\mathcal{I}} 1/|\hat{h}_b^m(\nu)|^2 d\nu,$$

with probability at least $1 - \delta$. Using also (14) we conclude that

$$\mathbb{E}\left[|\widehat{B}^m(\omega)|^2\right]\mathbb{E}\left[\frac{1}{|\widehat{B}^m(\omega)|^2}\right] = \int_{\mathcal{I}}|\hat{h}_b^m(\nu)|^2 d\nu \int_{\mathcal{I}} 1/|h_b^m(\nu)|^2 d\nu$$

is larger than $c/2$ also with probability $1 - \delta$ for sufficiently large $m$. As $c$ was chosen arbitrary large, this proves that

$$\int_{\mathcal{I}}|\hat{h}_b^m(\nu)|^2 d\nu \int_{\mathcal{I}} 1/|h_b^m(\nu)|^2 d\nu \underset{m\to+\infty}{\overset{p}{\to}} +\infty. \tag{18}$$

Hence the inverse of the above quantity converges to zero in probability, which shows that

$$\rho_{\mathbf{X}\to\mathbf{Y}}\rho_{\mathbf{Y}\to\mathbf{X}} = \frac{1}{\int_{\mathcal{I}}|\hat{h}_b^m(\nu)|^2 d\nu \int_{\mathcal{I}} 1/|h_b^m(\nu)|^2 d\nu} \underset{m\to+\infty}{\overset{p}{\to}} 0.$$

∎

### A.5. Proof of Theorem 2

The following lemmas will be helpful in proving Theorem 2.

**Lemma 3** *(Serre, 2010) For a given Hermitian matrix $H$ and any principal submatrix of $H$, $H'$, their spectral radius $\rho_s$ satisfies*

$$\rho_s(H) \geq \rho_s(H').$$

**Lemma 4** *(Gray, 2006) Let $f : [-\frac{1}{2}, \frac{1}{2}) \to \mathbb{R}$ such that $f \in L^1$ be a bounded function and suppose $t_k$ is its Fourier series coefficients, i.e.*

$$t_k = \int_{-\frac{1}{2}}^{\frac{1}{2}} f(\nu)e^{i2\pi k\nu} d\nu, \quad k \in \mathbb{Z}.$$

*Consider Toeplitz matrices $T_n$ defined as*

$$[T_n]_{ij} = t_{i-j} \quad i, j \in \{0, ..., n-1\}$$

*with eigenvalues $\tau_{n,k}(0 \leq k \leq n-1)$. Then if $(t_k)$ is absolutely summable we get:*

$$\min_{x\in[-\frac{1}{2},\frac{1}{2})} f(x) \leq \tau_{n,i} \leq \max_{x\in[-\frac{1}{2},\frac{1}{2})} f(x)$$

We also exploit a concentration of measure result for Lipschitz continuous functions of random elements of the special orthogonal group $SO(N) = \{M \in \mathbb{R}^{N\times N}, M^\top M = I_N, |M| = 1\}$.

**Lemma 5** *(Guionnet, 2009, Corollary 6.14) For any differentiable function $f : SO(N) \to \mathbb{R}$ such that, for any $X, Y \in SO(N), |f(X) - f(Y)| \leq K\|X - Y\|_F$, where $\|\|_2$ denotes the Forbenius norm, we have for any random matrix $O$ Haar distributed on $SO(N)$ and for all $\delta \geq 0$*

$$|f(O) - \mathbb{E}_O[f(O)]| < 4K\delta$$

*with probability at least $1 - 2e^{-N\delta^2}$*

K thus represents a Lipschitz constant of $f$. This concentration result implies another one more specific to our case.

**Lemma 6** *Let $A, B$ be square matrices in $\mathbb{R}^{n \times n}$, assuming $A$ is symmetric and let $O$ be a random rotation matrix, Haar distributed on $SO(n)$. We denote $\tau_n[.] = \frac{1}{n}\mathrm{tr}[.]$, $\|\|_F$ the Frobenius norm and $\|\|_2$ the spectral (operator) norm. For all $\delta \geq 0$*

$$\left|\tau_n(AOBO^T) - \tau_n(A)\tau_n(B)\right| < 8\|A\|_F\|B\|_2\delta$$

*with probability at least $1 - 2e^{-n^3\delta^2}$, and*

$$\left|\tau_n(AOBO^T) - \tau_n(A)\tau_n(B)\right| < 8\rho(A)\rho(B)\delta$$

*with probability at least $1 - 2e^{-n^2\delta^2}$.*

**Proof**

The proof follows two steps in order to apply the above lemma to the function $O \mapsto \tau_n(AOBO^T)$ restricted to $SO(n)$.

Step 1: First, we show $O \mapsto \tau_n(AOBO^T)$ is Lipschitz continuous on $SO(n)$. Let $U, V \in SO(n)$, then

$$\tau_n(AUBU^T) - \tau_n(AVBV^T) = \tau_n(A(U-V)BU^T) + \tau_n(AVB(U-V)^T)$$

applying the Cauchy-Schwartz inequality on the two Frobenius scalar products ($|\mathrm{Tr}(XY^T)| \leq \|X\|_F\|Y\|_F$) we get

$$\left|\tau_n(AUBU^T) - \tau_n(AVBV^T)\right| \leq \frac{1}{n}\|A\|_F \left(\|(U-V)BU^T\|_F + \|VB(U-V)^T\|_F\right)$$

Using the matrix norm inequality[3] $\|XY\|_F \leq \|X\|_F\|Y\|_2$ we get

$$\left|\tau_n(AUBU^T) - \tau_n(AVBV^T)\right| \leq \frac{1}{n}\|A\|_F\|U-V\|_F \left(\|BU^T\|_2 + \|VB\|_2\right)$$

and finally by rotation invariance of the spectral norm

$$\left|\tau_n(AUBU^T) - \tau_n(AVBV^T)\right| \leq \frac{2}{n}\|A\|_F\|B\|_2\|U-V\|_F$$

Step 2: Now we show $\mathbb{E}_O\tau_n(AOBO^T) = \tau_n(A)\tau_n(B)$. By linearity of the trace we have

$$\mathbb{E}_O\left[\tau_n(AOBO^T)\right] = \tau_n(A\mathbb{E}_O\left[OBO^T\right]).$$

It is easy to see that $\mathbb{E}_O[OBO^T]$ commutes with every $Q \in SO(n)$ due to $Q\mathbb{E}_0[OBO^T]Q^T = \mathbb{E}_O[OBO^T]$. Thus, $\mathbb{E}_0[OBO^T]$ is a multiple of the identity (due to Schur's lemma, otherwise $SO(n)$ would not act irreducibly on $\mathbb{R}^n$). Finally, we conclude $\mathbb{E}_0[OBO^T] = \tau_n(B)\mathbf{1}$ because $\tau_n(\mathbb{E}_0[OBO^T]) = \tau_n(B)$.

---

3. http://mathoverflow.net/questions/59918/submultiplicity-of-matrix-norm-is-ab-f-leq-a-2b-f

Finally, combining the results from both steps we can apply Lemma 5, we get, using the above Lipschitz constant $K = \frac{2}{n}\|A\|_F\|B\|_2$,

$$\left|\tau_n(AOBO^T) - \tau_n(A)\tau_n(B)\right| < \frac{8}{n}\|A\|_F\|B\|_2\delta$$

with probability at least $1 - 2e^{-n\delta^2}$, which gives the first bound of the lemma. The second bound is obtained using the matrix norm inequality $\|A\|_F \leq \sqrt{n}\|A\|_2$.

∎

Now we are ready to prove Theorem 2:

**Theorem 2 (Concentration of measure for FIR filters)** *Assume a length $m$ mechanism's impulse response* $h_{\mathbf{X}\to\mathbf{Y}}$ *with a spherically symmetric generative model and a fixed input PSD $S_{xx}$, then for any given $\varepsilon > 0$, with probability at least $1 - 2\exp(-m\varepsilon^2)$, we have*

$$|\rho_{\mathbf{X}\to\mathbf{Y}} - 1| \leq \frac{8\varepsilon}{\langle S_{xx}\rangle}\max_{\nu} S_{xx}(\nu).$$

**Proof** Without loss of generality and for the sake of simplicity we only consider the positive indices of the time series and we take the filter to be causal; other cases can be treated in a similar way. Then the following relation holds between input and output of the filter:

$$\forall i, \quad 0 \leq i \leq N-1 \quad Y_i = \sum_{j=0}^{m-1} b_j X_{i-j}$$

Formulated in terms of matrices the above relation can be represented as

$$\begin{bmatrix} Y_0 \\ Y_1 \\ \vdots \\ Y_{N-2} \\ Y_{N-1} \end{bmatrix} = H_{N,m} \begin{bmatrix} X_{-m+1} \\ X_{-m+2} \\ \vdots \\ X_{N-2} \\ X_{N-1} \end{bmatrix},$$

where $H_{N,m}$ is a $N \times (N + m - 1)$ matrix as follows

$$\begin{bmatrix} b_{m-1} & b_{m-2} & \cdots & b_0 & 0 & \cdots & 0 & 0 \\ 0 & b_{m-1} & \cdots & b_1 & b_0 & \cdots & 0 & 0 \\ & & \ddots & & & & & \\ 0 & 0 & \cdots & b_{m-1} & \cdots & b_1 & b_0 & 0 \\ 0 & 0 & \cdots & 0 & b_{m-1} & \cdots & b_1 & b_0 \end{bmatrix}$$

We define $\Sigma_X^i \in M_{m\times m}(\mathbb{R})$ to be the covariance matrices as follows:

$$\forall i, \quad 0 \leq i \leq N-1, \ \forall(j,k), \ 0 \leq j,k \leq m-1, \quad [\Sigma_X^i]_{jk} = \mathbb{C}\mathrm{ov}(X_{i+j}, X_{i+k}) \quad (19)$$

Since the time series under consideration are weakly stationary, $\Sigma_X^i$ is independent of $i$ and we denote it $\Sigma_m$. If we take $\Sigma_{X_{0:N-1}}, \Sigma_{Y_{0:N-1}} \in M_{N \times N}(\mathbb{R})$ to be the covariance matrices for $X_{0:N-1}$ and $Y_{0:N-1}$ respectively, then we have

$$\Sigma_{Y_{0:N-1}} = H_{N,m} \Sigma_{X_{-m+1:N-1}} H_{N,m}^\top \, .$$

where $\Sigma_{X_{i:j}}$ is the covariance matrix of $X_{i:j}$ which is a principal submatrix of $\Sigma_X$. We also define $\Sigma_{Y_{0:N-1}}^U$ to be the covariance matrix of the output for FIR $\mathcal{S}'$ with $\mathbf{b}' = U^\top \mathbf{b}$. Furthermore assume the spectrum of the output for this filter is $S_{yy}^U$. One can write the diagonal elements of $\Sigma_{Y_{0:N-1}}^U$ based on the above equation as follows:

$$[\Sigma_{Y_{0:N-1}}]_{ii} = \mathbf{b}^\top \Sigma \mathbf{b},$$

which is therefore equal to the normalized trace of $\Sigma_{Y_{0:N-1}}$.

Due to the spherical invariance assumption, $\mathbf{b}$ is a single sample from the product $R\mathbf{U}$ (Bryc, 2012, Theorem 4.1.2). Moreover, $\mathbf{U}$ can obviously be rewritten as the first column of a random rotation matrix, $\mathbf{U} = O\mathbf{x_0}$, where $x_0$ is the canonical basis vector with first component 1, while other components are zero, and $O$ is a random matrix Haar distributed on SO(m).

Taking $A = \mathbf{x_0}\mathbf{x_0}^\top$ and $B = \Sigma_m$ (defined in (19)) in Lemma 6 we get

$$|\tau_m \left( \mathbf{x_0}\mathbf{x_0}^\top O^\top \Sigma_m O \right) - \tau_m(\Sigma_m)\tau_m \left( \mathbf{x_0}\mathbf{x_0}^\top \right) | \leq 8\delta \|\Sigma_m\|_2 \langle \mathbf{x_0}, \mathbf{x_0} \rangle^{1/2} \tag{20}$$

with probability at least $1 - 2e^{-m^3\delta^2}$.

Replacing $\mathbf{b}$ by its corresponding random variable, we get

$$\mathcal{P}(\mathbf{Y}) = \tau_N(\Sigma_{Y_{0:N-1}}) = R^2 \mathbf{x_0}^\top O^\top \Sigma_m O \mathbf{x_0} \, ,$$

such that

$$\mathcal{P}(\mathbf{Y}) = R^2 \mathrm{tr}[\mathbf{x_0}^\top O^\top \Sigma_m O \mathbf{x_0}] = R^2 m \tau_m [\mathbf{x_0}\mathbf{x_0}^\top O^\top \Sigma_m O] \, .$$

Moreover, we notice that according to the generative model, the squared norm of the impulse response is sampled from the random variable $R^2$, hence (20) becomes

$$| \frac{\mathcal{P}(\mathbf{Y})}{\|h\|^2} / m - \tau_m(\Sigma_m)/m | \leq 8\delta \|\Sigma_m\|_2 \tag{21}$$

with the same probability as above.

On the other hand the elements of diagonals of $\Sigma_m$ are $C_X(0)$. Therefore:

$$\tau_m(\Sigma) = \mathcal{P}(\mathbf{X}).$$

Since $\Sigma$ is a principal submatrix of $\Sigma_{X_{0:N-1}}$, by Lemma 3

$$\|\Sigma_m\|_2 \leq \rho(\Sigma_{X_{0:N-1}}).$$

Because $C_X(\tau)$'s are absolutely summable, based on Lemma 4 we get

$$\rho(\Sigma_{X_{0:N-1}}) \le \max_\nu S_{xx}(\nu),$$

and then inequality (21) can be rewritten as

$$\left| \frac{\mathcal{P}(\mathbf{Y})}{\|h\|^2 \mathcal{P}(\mathbf{X})} - 1 \right| \le 8m\delta \frac{\max_\nu S_{xx}(\nu)}{\mathcal{P}(\mathbf{X})}$$

occurring with probability at least $1 - 2e^{-m^3\delta^2}$. As a consequence

$$\left| \frac{\mathcal{P}(\mathbf{Y})}{\|h\|^2 \mathcal{P}(\mathbf{X})} - 1 \right| \le 8\delta \frac{\max_\nu S_{xx}(\nu)}{\mathcal{P}(\mathbf{X})}$$

occurs with probability at least $1 - 2e^{-m\delta^2}$, which completes the proof. ∎

### A.6. Proof for Proposition 2

First, we derive to helpful lemmas that are used to prove the proposition; the first one being used to prove the first part and the second lemma is used to infer the second part of the proposition above.

**Lemma 7** *For $f \in L^2(\mathcal{I})$ non-constant, such that $1/f \in L^2(\mathcal{I})$, we have*

$$\int_\mathcal{I} f(x)^2 dx \cdot \int_\mathcal{I} \frac{1}{f(x)^2} dx > 1$$

**Proof** The inequality follows from the Cauchy-Schwartz inequality for the scalar product on $L^2(\mathcal{I})$, stating

$$\left| \int_\mathcal{I} a(x)b(x)dx \right|^2 < \int_\mathcal{I} a(x)^2 dx \cdot \int_\mathcal{I} b(x)^2 dx \,,$$

for non co-linear elements $a$ and $b$ of $L^2(\mathcal{I})$. As colinearity of $a = f^2$ and its inverse $b$ would imply they are constant, we get the result immediately. ∎

**Lemma 8** *Let $f \in L^1(\mathcal{I})$ be positive, non-constant, such that $1/f \in L^1(\mathcal{I})$ and $\int_\mathcal{I} f(x)dx = 1$. Assume $\exists \alpha > 0, \forall x \in \mathcal{I}, f(x) \le 2 - \alpha$, then*

$$\int_\mathcal{I} f(x)dx \cdot \int_\mathcal{I} \frac{1}{f(x)} dx \ge 1 + \alpha \int_\mathcal{I} (f(x) - 1)^2 dx$$

**Proof** We denote $s(x) = f(x) - 1$. Then $\int_\mathcal{I} s(x)dx = 0$ and

$$\int_\mathcal{I} f(x)dx . \int_\mathcal{I} \frac{1}{f(x)} dx - 1 = \int_\mathcal{I} \frac{-s(x)}{1 + s(x)} dx$$

For $x > -1$, we have

$$\frac{-x}{1+x} \ge x^2 - x^3 - x. \tag{22}$$

Replacing $s(x)$ with $x$ in (22) we get:

$$\int_{\mathcal{I}} f(x)dx \cdot \int_{\mathcal{I}} \frac{1}{f(x)}dx - 1 \geq \int_{\mathcal{I}} s(x)^2(1 - s(x))dx.$$

Since $1 - s(x) = 2 - f(x) \geq \alpha > 0$,

$$\int_{\mathcal{I}} f(x)dx \cdot \int_{\mathcal{I}} \frac{1}{f(x)}dx - 1 \geq \alpha \int_{\mathcal{I}} s(x)^2 dx$$

∎

Now we can prove Proposition 2.

**Proposition 2 (Forward-backward inequality)** *For a given linear filter $\mathcal{S}$ with impulse response* $\mathbf{h}_{\mathbf{X}\to\mathbf{Y}}$*, input PSD $S_{xx}$ and output PSD $S_{yy}$, let $\mathcal{S}$ have a non-constant modulus frequency response* $\widehat{\mathrm{h}}_{\mathbf{X}\to\mathbf{Y}}$*, and assume there exists $\alpha > 0$ such that for all $\nu \in \mathcal{I}$,*

$$|\widehat{\mathrm{h}}_{\mathbf{X}\to\mathbf{Y}}(\nu)|^2 \leq (2 - \alpha)\left\langle |\widehat{\mathrm{h}}_{\mathbf{X}\to\mathbf{Y}}|^2 \right\rangle.$$

*Then*

$$\rho_{\mathbf{X}\to\mathbf{Y}} \cdot \rho_{\mathbf{Y}\to\mathbf{X}} \leq \left[1 + \alpha CV\left(\left(|\widehat{\mathrm{h}}_{\mathbf{X}\to\mathbf{Y}}|^2\right)^2\right)\right]^{-1} < 1. \tag{11}$$

**Proof**

(i) Using the definition of Spectral Dependency Ratios and Lemma 7 it easily follows that

$$\rho_{\mathbf{X}\to\mathbf{Y}}\rho_{\mathbf{Y}\to\mathbf{X}} = \frac{1}{\langle |\widehat{\mathbf{h}}_{\mathbf{X}\to\mathbf{Y}}|^2\rangle\langle 1/|\widehat{\mathbf{h}}_{\mathbf{X}\to\mathbf{Y}}|^2\rangle} < 1$$

(ii) Applying Lemma 8 to

$$f = |\widehat{\mathbf{h}}_{\mathbf{X}\to\mathbf{Y}}|^2 / \int_{\mathcal{I}} |\widehat{\mathbf{h}}_{\mathbf{X}\to\mathbf{Y}}|^2 = |\widehat{\mathbf{h}}_{\mathbf{X}\to\mathbf{Y}}|^2 / \|\mathbf{h}_{\mathbf{X}\to\mathbf{Y}}\|_2^2,$$

we get inequality (11).

∎

### A.7. Proof of Theorem 3

**Proof** Lemma 1 implies for any fixed $\epsilon > 0$

$$P(|\rho_{\mathbf{X}\to\mathbf{Y}} \cdot \rho_{\mathbf{Y}\to\mathbf{X}}| > \epsilon) \underset{m\to+\infty}{\to} 0$$

as the dimension of the filter increases. Moreover, Theorem 2 implies

$$P(|\rho_{\mathbf{X}\to\mathbf{Y}}| < 1 - \epsilon) \underset{m\to+\infty}{\to} 0.$$

Given that

$$P(|\rho_{\mathbf{Y}\to\mathbf{X}}| < 1/2) \geq P(\{\rho_{\mathbf{Y}\to\mathbf{X}}\cdot\rho_{\mathbf{X}\to\mathbf{Y}} < 1/4\}\&\{\rho_{\mathbf{X}\to\mathbf{Y}} > 1/2\})$$
$$= 1 - P(\rho_{\mathbf{Y}\to\mathbf{X}}\cdot\rho_{\mathbf{X}\to\mathbf{Y}} \geq 1/4) - P(\rho_{\mathbf{X}\to\mathbf{Y}} \leq 1/2),$$

$P(|\rho_{\mathbf{Y}\to\mathbf{X}}| < 1/2)$ thus tends to one as $m \to +\infty$. Moreover

$$P(\rho_{\mathbf{Y}\to\mathbf{X}} < \rho_{\mathbf{X}\to\mathbf{Y}}) \geq P(\{|\rho_{\mathbf{X}\to\mathbf{Y}}| > 1/2)\}\&\{|\rho_{\mathbf{Y}\to\mathbf{X}}| < 1/2\})$$
$$= 1 - P(\{|\rho_{\mathbf{X}\to\mathbf{Y}}| \leq 1/2)\} \text{ or } \{|\rho_{\mathbf{Y}\to\mathbf{X}}| \geq 1/2\})$$
$$\geq 1 - P(|\rho_{\mathbf{X}\to\mathbf{Y}}| \leq 1/2) - P(|\rho_{\mathbf{Y}\to\mathbf{X}}| \geq 1/2),$$

such that $P(\rho_{\mathbf{Y}\to\mathbf{X}} < \rho_{\mathbf{X}\to\mathbf{Y}})$ tends to one a $m$ tends to infinity. ∎

## A.8. Proof of Theorem 4

For convenience we restate the theorem here:

**Theorem 4** *Assume the forward generative model of the previous section, and $(\widetilde{\mathbf{X}}, \widetilde{\mathbf{Y}})$ the time series resulting from the decimation of this model by an integer factor $D$ using an ideal anti-aliasing filter. Then for any given $\epsilon$ with $0 < \epsilon < \frac{1}{4D}$ we have*

$$|\rho_{\widetilde{\mathbf{X}}\to\widetilde{\mathbf{Y}}} - 1| \leq \epsilon\left(K + (1 + \epsilon K)\frac{2}{1 - 4D\epsilon}\right),$$

*with probability $\delta := (1 - \exp(-\kappa(m-1)\epsilon^2))^2$, where $\kappa$ is a positive global constant (independent of $m$, and $\epsilon$) and $K = \frac{\max_{|\nu|<1/2D} S_{xx}(\nu)}{\int_0^{1/2D} S_{xx}(\nu)d\nu}$.*

Let us first write the forward SDR of the downsampled data:

$$\rho_{\widetilde{\mathbf{X}}\to\widetilde{\mathbf{Y}}} := \frac{\langle S_{\widetilde{y}\widetilde{y}}\rangle}{\langle S_{\widetilde{x}\widetilde{x}}\rangle\langle S_{\widetilde{y}\widetilde{y}}/S_{\widetilde{x}\widetilde{x}}\rangle} = \frac{2/D\int_0^{1/2} S_{yy}(\nu/D)d\nu}{\left(2/D\int_0^{1/2} S_{xx}(\nu/D)d\nu\right)\left(2\int_0^{1/2} S_{yy}(\nu/D)/S_{xx}(\nu/D)\,d\nu\right)}$$

$$\rho_{\widetilde{\mathbf{X}}\to\widetilde{\mathbf{Y}}} := \frac{\langle S_{\widetilde{y}\widetilde{y}}\rangle}{\langle S_{\widetilde{x}\widetilde{x}}\rangle\langle S_{\widetilde{y}\widetilde{y}}/S_{\widetilde{x}\widetilde{x}}\rangle} = \frac{\int_0^{1/2D} S_{yy}(\nu)d\nu}{\left(\int_0^{1/2D} S_{xx}(\nu)d\nu\right)\left(2D\int_0^{1/2D} |\widehat{\mathbf{h}}(\nu)|^2\,d\nu\right)}$$

Second, we can apply the previous concentration of measure result of (Shajarisales et al., 2015) (Theorem 1) to the low pass filtered causes and effects as they satisfy the requirements of the generative model, then the corresponding forward SDR is

$$\rho_{a*\mathbf{X}\to a*\mathbf{Y}} = \frac{\int_0^{1/2D} S_{yy}(\nu)d\nu}{\left(\int_0^{1/2D} S_{xx}(\nu)d\nu\right)\left(2\int_0^{1/2} |\widehat{\mathbf{h}}(\nu)|^2\,d\nu\right)},$$

and we have

$$|\rho_{a*\mathbf{X}\to a*\mathbf{Y}} - 1| \leq 2\epsilon\frac{\max_{|\nu|<1/2D} S_{XX}(\nu)}{\int_0^{1/2D} S_{xx}(\nu)d\nu}.$$

with probability at least $1 - \exp(-\kappa(m-1)\varepsilon^2)$.

Moreover, we have (using $\mathbf{h} = \mathbf{Ub}$ and $\mathbf{U^H U} = \mathbf{I_m}$):

$$\rho_{\widetilde{\mathbf{X}} \to \widetilde{\mathbf{Y}}} = \rho_{a*\mathbf{X} \to a*\mathbf{Y}} \cdot \frac{\int_0^{1/2} |\widehat{\mathbf{h}}(\nu)|^2 \, d\nu}{D \int_0^{1/2D} |\widehat{\mathbf{h}}(\nu)|^2 \, d\nu} = \rho_{a*\mathbf{X} \to a*\mathbf{Y}} \cdot \left| \frac{\|\mathbf{b}\|^2}{2D \int_0^{1/2D} |\widehat{\mathbf{h}}(\nu)|^2 \, d\nu} \right|$$

As a consequence:

$$|\rho_{\widetilde{\mathbf{X}} \to \widetilde{\mathbf{Y}}} - 1| \leq |\rho_{a*\mathbf{X} \to a*\mathbf{Y}} - 1| + \rho_{a*\mathbf{X} \to a*\mathbf{Y}} \left| \frac{\|\mathbf{b}\|^2}{2D \int_0^{1/2D} |\widehat{\mathbf{h}}(\nu)|^2 \, d\nu} - 1 \right|$$

Hence the bound:

$$|\rho_{\widetilde{\mathbf{X}} \to \widetilde{\mathbf{Y}}} - 1| \leq 8\epsilon \frac{\max_{|\nu| < 1/2D} S_{XX}(\nu)}{\int_0^{1/2D} S_{xx}(\nu) d\nu} + \left( 1 + 8\epsilon \frac{\max_{|\nu| < 1/2D} S_{XX}(\nu)}{\int_0^{1/2D} S_{xx}(\nu) d\nu} \right) \left| \frac{\|\mathbf{b}\|^2}{2D \int_0^{1/2D} |\widehat{\mathbf{h}}(\nu)|^2 \, d\nu} - 1 \right|$$

Only the rightmost term remains to be bounded. We will thus proceed to the evaluation of $\int_0^{1/2D} |\widehat{\mathbf{h}}(\nu)|^2 \, d\nu$; first by estimating the integral using a left point approximation using a grid of size $M$ (we choose M as a multiple of $2D$):

$$L_M^D(\widehat{\mathbf{h}}) = \frac{1}{M} \sum_{k=0}^{(M/2D)-1} |\widehat{\mathbf{h}}(k/M)|^2$$

By using a bound on the derivate of $\widehat{\mathbf{h}}$ ($|\widehat{\mathbf{h}}'| \leq \sqrt{m}^3 2\pi \|\mathbf{b}\|$) we can bound the left point approximation:

$$\left| L_M^D(\widehat{\mathbf{h}}) - \int_0^{1/2D} |\widehat{\mathbf{h}}(\nu)|^2 \, d\nu \right| \leq \frac{\sqrt{m}^3 2\pi \|\mathbf{b}\|}{(2D)^2 2M}$$

Now $L_M^D(\widehat{\mathbf{h}})$ can be evaluated using the concentration of measure theorem from (Janzing et al. 2010). Let $\mathbf{F}_{M,m}$ be the matrix implementing the M frequency points discrete Fourier Transform (DFT) of an m-time points vector[4] such that $\mathbf{F}_{M,m} = \{\exp -2\mathbf{i}\pi kn/M\}_{k=0..M-1;n=0..m-1}$, and let $\mathbf{L}_{M,D}$ the diagonal matrix such that $(\mathbf{L}_{M,D})_{kk} = \mathbb{1}_{k < M/(2D)} ()$, we have

$$M \cdot L_M^D(\widehat{\mathbf{h}}) = (\mathbf{F}_{M,m} \mathbf{Ub})^H \mathbf{L}_{M,D} \mathbf{F}_{M,m} \mathbf{Ub} = \mathbf{b}^T \mathbf{U}^T \mathbf{F}_{M,m}^H \mathbf{L}_{M,D} \mathbf{F}_{M,m} \mathbf{Ub}$$

As a consequence, we get the following concentration of measure result, with probability $\delta = 1 - \exp(-\kappa(m-1)\epsilon^2)$

$$\left| M \cdot L_M^D(\widehat{\mathbf{h}}) - \mathbf{b}^T \mathbf{b} \frac{1}{m} \mathbf{Tr}(\mathbf{F}_{M,m}^H \mathbf{L}_{M,D} \mathbf{F}_{M,m}) \right| \leq 2\epsilon \|\mathbf{b}\|^2 \left\| \mathbf{F}_{M,m}^H \mathbf{L}_{M,D} \mathbf{F}_{M,m} \right\|_o,$$

where $\|\cdot\|_o$ denotes the operator norm, which can be bounded as follows:

$$\left\| \mathbf{F}_{M,m}^H \mathbf{L}_{M,D} \mathbf{F}_{M,m} \right\|_o \leq \|\mathbf{F}_{M,m}\|_o^2 \|\mathbf{L}_{M,D}\|_o$$

---

4. this can be seen as padding this input vector with zeros to get a N-dimensional vector and appliying the classical NxN DFT matrix

Moreover $\|\mathbf{L}_{M,D}\|_o = 1$ and the operator norm of the zero padded FFT matrix can be bounded by writing it down using the full M-dimentional FFT matrix, such that $\mathbf{F}_{M,m} = \mathbf{F}_{M,M}\begin{bmatrix} \mathbf{I}_m \\ \mathbf{0} \end{bmatrix}$

$$\|\mathbf{F}_{M,m}\|_o \leq \|\mathbf{F}_{M,M}\|_o \left\|\begin{bmatrix} \mathbf{I}_m \\ \mathbf{0} \end{bmatrix}\right\|_o \leq \sqrt{M}$$

As a consequence we get the bound (noticing $\mathbf{Tr}(\mathbf{F}_{M,m}^H \mathbf{L}_{M,D}\mathbf{F}_{M,m}) = mM/(2D)$):

$$|M \cdot L_M^D(\widehat{\mathbf{h}}) - \|\mathbf{b}\|^2 M/(2D)| \leq 2\epsilon\|\mathbf{b}\|^2 M,$$

such that

$$|L_M^D(\widehat{\mathbf{h}}) - \frac{\|\mathbf{b}\|^2}{2D}| \leq 2\epsilon\|\mathbf{b}\|^2$$

We note that this bound does not depend on $M$, such that by taking the limit $M \to \infty$ we can bound $\int_0^{1/2D} |\widehat{\mathbf{h}}(\nu)|^2 \, d\nu$:

$$\left|\frac{\|\mathbf{b}\|^2}{2D} - \int_0^{1/2D} |\widehat{\mathbf{h}}(\nu)|^2 \, d\nu\right| \leq 2\epsilon\|\mathbf{b}\|^2.$$

Putting together all bounds we get

$$|\rho_{\widetilde{\mathbf{X}}\to\widetilde{\mathbf{Y}}} - 1| \leq \epsilon \frac{\max_{|\nu|<1/2D} S_{xx}(\nu)}{\int_0^{1/2D} S_{xx}(\nu)d\nu} + \left(1 + \epsilon\frac{\max_{|\nu|<1/2D} S_{xx}(\nu)}{\int_0^{1/2D} S_{xx}(\nu)d\nu}\right)\frac{2\epsilon\|\mathbf{b}\|^2}{2D\int_0^{1/2D}|\widehat{\mathbf{h}}(\nu)|^2 \, d\nu},$$

and using again the previous bound we finally get:

$$|\rho_{\widetilde{\mathbf{X}}\to\widetilde{\mathbf{Y}}} - 1| \leq \epsilon \frac{\max_{|\nu|<1/2D} S_{xx}(\nu)}{\int_0^{1/2D} S_{xx}(\nu)d\nu} + \left(1 + \epsilon\frac{\max_{|\nu|<1/2D} S_{xx}(\nu)}{\int_0^{1/2D} S_{xx}(\nu)d\nu}\right)\frac{2\epsilon}{1 - 4D\epsilon}$$

with probability $\delta^2 = (1 - \exp(-\kappa(m-1)\epsilon^2))^2$ and for $\epsilon < \frac{1}{4D}$.

## Appendix B. Additional background

### B.1. Fourier transform of sequences

Consider a sequence of real or complex numbers $\mathbf{a} = \{a_t, t \in \mathbb{Z}\}$ as a deterministic vector from the $\ell^1(\mathbb{Z})$ sequence space of bounded $\ell^1$ norm, $\|\mathbf{a}\|_1 = \sum_{t\in\mathbb{Z}} |a_t| < \infty$. The support of the sequence is the subset $\text{Supp}(\mathbf{a}) = \{k \in \mathbb{Z}, |a_k| > 0\} \subset \mathbb{Z}$. Its Discrete-Time Fourier Transform (DTFT) is defined as

$$\widehat{a}(\nu) = \sum_{t\in\mathbb{Z}} a_t \exp(-\mathbf{i}2\pi\nu t), \, \nu \in \mathbb{R}$$

Note that the DTFT of such sequence is a continuous 1-periodic function of the normalized frequency $\nu$ and can be characterized by its values on any unit length interval (Vetterli et al., 2014, Chapter 3). We will use the unit interval centered around zero $[-1/2, 1/2] =: \mathcal{I}$. In case the original sequence is real-valued, its Fourier transform is conjugate symmetric ($\widehat{a}(-\nu) = \widehat{a}^*(\nu)$) such that its modulus is an even function. This will be the case in the remainder of this paper. The squared $\ell^2$-norm $\|\mathbf{a}\|_2^2 = \sum_t |a_t|^2$ is often called *energy*. By Parseval's theorem, it can be expressed in the Fourier domain by $\|\mathbf{a}\|_2^2 = \int_{-1/2}^{1/2} |\widehat{a}(\nu)|^2 d\nu$. To simplify notations, we will denote by $\langle . \rangle$ the integral (or average) of a function over the unit interval $\mathcal{I}$, such that $\|\mathbf{a}\|_2^2 = \langle|\widehat{a}|^2\rangle$.

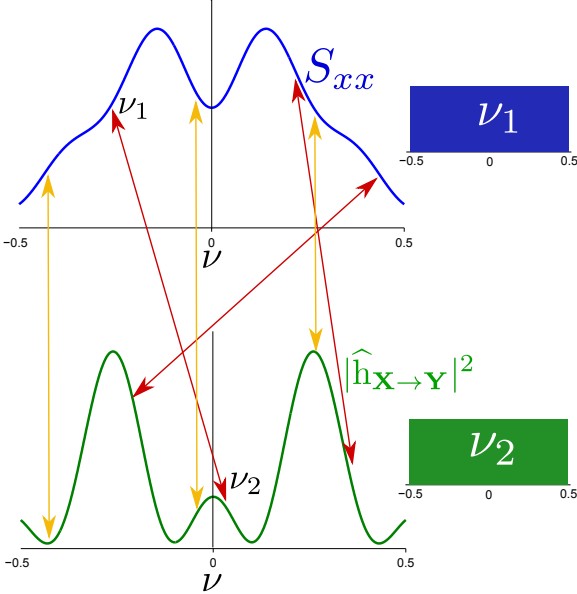

Figure 5: Probabilistic view of spectral independence: the product has the same expectation when frequencies are matched at random (red), compared to when they are chosen identical (orange).

### B.2. Probabilistic interpretation

To motivate why eq. (3) is called an *independence* condition, we notice that the difference between the left and the right hand side can be written as a covariance:

$$\langle S_{xx} \cdot |\widehat{\mathrm{h}}_{\mathbf{X} \to \mathbf{Y}}|^2 \rangle - \langle S_{xx} \rangle \langle |\widehat{\mathrm{h}}_{\mathbf{X} \to \mathbf{Y}}|^2 \rangle = \mathbb{C}\mathrm{ov}\left(S_{xx}, |\widehat{\mathrm{h}}_{\mathbf{X} \to \mathbf{Y}}|^2\right) . \tag{23}$$

To understand the rephrasing as *covariance*, note that the maps $\nu \mapsto S_{xx}(\nu)$ and $\nu \mapsto |\widehat{\mathrm{h}}_{\mathbf{X} \to \mathbf{Y}}(\nu)|^2$ can be formally interpreted as random variables on the probability space $[-1/2, 1/2]$ equipped with the uniform measure. From the purely mathematical point of view, this interpretation is certainly justified because any measurable map from a measure space to $\mathbb{R}$ can be considered as a random variable. For a more intuitive approach, one can think of a random experiment where one chooses a frequency $\nu \in [-1/2, 1/2]$ according to the uniform distribution and then observes $S_{xx}(\nu)$ and $|\widehat{\mathrm{h}}_{\mathbf{X} \to \mathbf{Y}}(\nu)|^2$. Finally, after a large (actually infinite) number of runs, the covariance of these quantities is given by (23). Accordingly, SIC can be interpreted as follows: The expectation of $S_{xx}(\nu)|\widehat{\mathrm{h}(\nu)}_{\mathbf{X} \to \mathbf{Y}}|^2$ is the same as if two frequencies $\nu_1$ and $\nu_2$ were drawn independently (as illustrated on Fig. 5) for both functions, which amounts to match at random the frequencies of the amplification factor and of the input PSD. Conversely, SIC would be violated, for instance, if the system mainly amplifies the frequencies with low intensity, as it happens for the 'anti-causal' direction in our motivating example in Subsection B.3. However, spectral independence does not correspond to classical statistical independence for two reasons. First, "independence" is measured at the level of the parameters of the causal system (filter coefficients and input PSD properties) and not at the level of the observed random variables. Second, statistical independence is a stronger statement than uncorrelatedness, and SIC corresponds essentially to the latter.

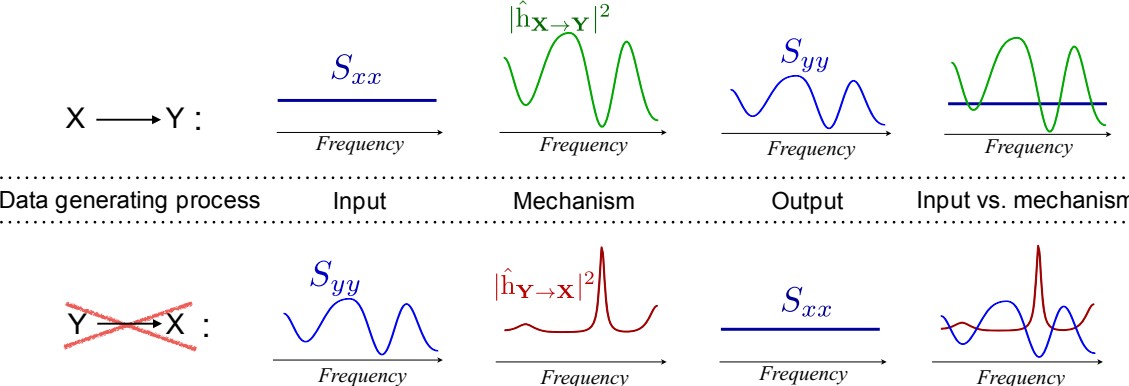

Figure 6: This figure shows two different possible causal scenarios: 1 - Upper Diagram where $\mathbf{X}$ causes $\mathbf{Y}$ and $\mathbf{Y}$ with PSD $S_{yy}$ is generated from $\mathbf{X}$ with $S_{xx}$ that gets multiplied pointwise with $|\mathbf{h}_{\mathbf{X} \to \mathbf{Y}}|^2$. 2 - Lower diagram where the inverse process takes place, where $S_{yy}$ is the PSD of input process.

### B.3. Intuitive Example

As illustrated in Fig. 6, assume that $\mathbf{X}$ is a white noise process, such that it has a spectral density function with uniform distribution, i.e. $S_{xx}(\nu) = c > 0$ for all $\nu \in [-1/2,\, 1/2]$. Here, we reject the hypothesis that $\mathbf{Y}$ causes $\mathbf{X}$ because it would require the corresponding filter mapping $\mathbf{Y}$ to $\mathbf{X}$ to be *tuned* relative to the input signal $\mathbf{Y}$: it amplifies the frequencies having small power in the input signal while it weakens those that have large power 'in order' to generate an output whose power is uniformly distributed over the frequencies. This results in $|\widehat{\mathbf{h}}_{\mathbf{Y} \to \mathbf{X}}|^2$ and $S_{yy}$ being strongly anticorrelated (see Fig. 6 bottom right panel). Thus, we have an extreme example of violation of ICM: $S_{yy}$ encodes all the second order statistics of $\mathbf{Y}$, the hypothetical input (for zero mean Gaussian time series it even describes the statistics of the input signal completely). On the other hand, $\widehat{\mathbf{h}}_{\mathbf{Y} \to \mathbf{X}}$ is a representative feature of the mechanism. Thus, the mechanism that generates $\mathbf{X}$ from $\mathbf{Y}$ is *informative* about its input.

### Appendix C. SIC and group invariance

**Group invariance perspective on ICM**  The group invariance framework assesses ICM by quantifying the *genericity* of the relationship between input $x$, corresponding to the PSD of the putative cause time series in the context of this paper, and mechanism $m$, corresponding to the filter. As explained in (Besserve et al., 2018), this requires defining two objects: (1) the *generic group* $\mathcal{G}$ is a compact topological group that acts on cause properties (i.e. elements of the group are transformation that modify the properties of the cause), thus equipped with a unique Haar probability measure $\mu_{\mathcal{G}}$, (2) the *contrast* $C$ is a real valued function.

The contrast and generic group introduced in such a way allow to compute the expected contrast value when randomly "breaking" the cause-mechanism relationship using generic transformations according to the following definition.

**Defnition 1** *Given a contrast $C$, the Expected Generic Contrast (EGC) of a cause mechanism pair $(x, m)$ is defined as:*

$$\langle C \rangle_{m,x} = \mathbb{E}_{g \sim \mu_{\mathcal{G}}} C(mgx)\,. \tag{24}$$

*The relation between $m$ and $x$ is $\mathcal{G}$-generic under $C$, whenever*

$$C(mx) = \langle C \rangle_{m,x} \tag{25}$$

*holds approximately.*

Equation (25) is the *genericity equation*, which expresses the idealized ICM postulate (hence the term "holds approximately"). Besserve et al. (2018) give a probabilistic interpretation of the concept of genericity. Assume we are given a generative model such that the cause $x$ is a single sample drawn from a meta-distribution[5] $\mathcal{P}_{\mathcal{X}}$ (see Fig. 7). To estimate genericity irrespective of the possible values of $x$, we consider the *genericity ratio* $C(mx)/\langle C \rangle_{m,x}$: this quantity should be close to one with high probability in order to satisfy ICM assumptions. Assume $\mathcal{P}_{\mathcal{X}}$ is a $\mathcal{G}$-invariant distribution, under mild assumptions (Wijsman, 1967) $x$ can be parametrized as $x = g\tilde{x}$ where $g$ is a sample from a $\mu_{\mathcal{G}}$-distributed variable $G$, and $\tilde{x}$ is a sample from anther RV independent of $G$.

$$\mathbb{E}_x\left[\frac{C(mx)}{\langle C \rangle_{m,x}}\right] = \mathbb{E}_{\tilde{x}}\mathbb{E}_g\left[\frac{C(mg\tilde{x})}{\langle C \rangle_{m,g\tilde{x}}}\right] = 1 \tag{26}$$

This tells us that the postulate of genericity is valid at least "on average" for the generative model. On the contrary, if this average would be different from 1 as it may happen for a non-invariant $\mathcal{P}_{\mathcal{X}}$, the postulate is unlikely be valid for individual examples. As represented on Fig. 7, the same reasoning can be applied when sampling the mechanism from an invariant distribution.

**The case of SIC.** By using the power of the time series $\mathbf{Y}$ in (1) as a contrast, and frequency translations as the generic group, we can show that Spectral Independence postulate correponds to a genericity equation in the group invariance framework.

**Proposition 4** *Let $\mathcal{G}$ be the group of modulo 1/2 translations that acts on the PSD by shifting its graph for positive frequencies ($\nu \in [0, 1/2]$) while the graph for negative frequencies is defined so that the transformed PSD is even. Using the total power as a contrast, $\mathcal{G}-$genericity is equivalent to SIC.*

**Proof** Suppose that for a given mechanism $m$ and given input $S_{xx}$ the $\mathcal{G}-$genericity assumption is satisfied. Noticing that $\mu_{\mathcal{G}}$ is the uniform probability measure over $[0, 1/2]$. This amounts to

$$\int_{-1/2}^{1/2} S_{xx}(\nu)|\widehat{\mathrm{h}}(\nu)|^2 d\nu = \int_0^{1/2}\left(2\int_0^{1/2}|\widehat{\mathrm{h}}(\nu)|^2 S_{xx}(\nu - g)\mu_G(g)d\nu\right)dg$$

$$= 4\int_0^{1/2}\int_0^{1/2}|\widehat{\mathrm{h}}(\nu)|^2 S_{xx}(\nu - g)d\nu dg$$

$$= 4\int_0^{1/2}|\widehat{\mathrm{h}}(\nu)|^2\left(\int_0^{1/2} S_{xx}(\nu - g)dg\right)d\nu$$

$$= \int_{-1/2}^{1/2} S_{xx}(\nu)d\nu \cdot \int_{-1/2}^{1/2}|\widehat{\mathrm{h}}(\nu)|^2 d\nu$$

This corresponds to the formula of the SIC postulate. $\square$ ∎

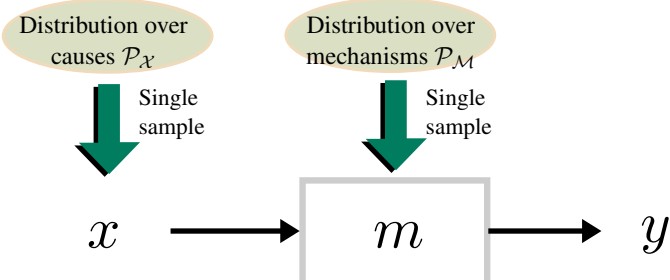

Figure 7: Generative model including distributions over causes and mechanisms. Modified from Besserve et al. (2018).

**Whitening: a way to enforce group invariance.** As can be understood from eq. (26) one possible source of misfit between ICM based approaches and real data is a lack of $\mathcal{G}$-invariance of the actual generating mechanism behind the observed data. In this section, we suggest a simple approach to adapt ICM methods to "baseline" properties of the set of causes. The genericity equation is compatible with a $\mathcal{G}$-invariant probabilistic generative model of the cause. Such an invariance assumption can be checked on real data, for example by verifying the $\mathcal{G}$-invariance of the empirical average of all putative causes. We can then seek to correct any lack of invariance: assume $b$ the average of empirically observed causes is not group invariant. If there exists an invertible transformation $w$ such that $w \circ b = u$ becomes invariant, we can define a new group invariance framework with

$$w^{-1} \circ \mathcal{G} \circ w = \{w^{-1} g w, \ g \in \mathcal{G}\},$$

as new generic group and $C_w : x \mapsto C(w \circ x)$ as a new contrast. $w$ is called a whitening transformation as it maps a predefined average distribution into a "white" invariant average distribution.

---

5. meta-distributions have some similarities with the approach of Lopez-Paz et al. (2015)

## Appendix D. Supplemental figures

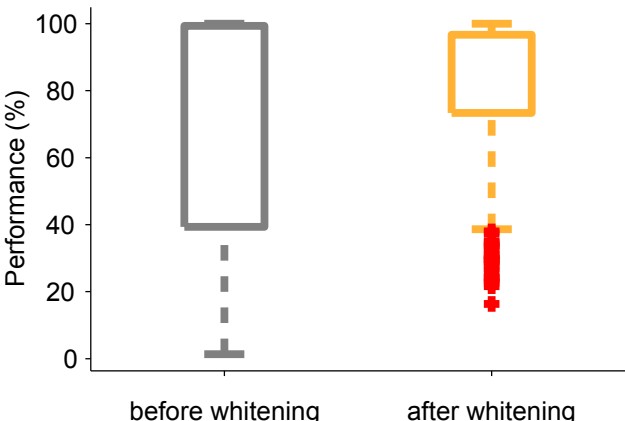

Figure 8: SIC performance comparison before and after whitening (see Sec. 7)

