# OpenReview forum: "Cause-effect inference through spectral independence in linear dynamical systems: theoretical foundations"
_cclear.cc/CLeaR/2022/Conference — CLeaR 2022 Poster_

### Official Review · Reviewer_P6NT · 2021-11-23

**Confidence:** 5
**Overall Score:** 9

**Main Review:**

The authors take the SIC-based causal discovery framework for time series and proves that under certain generative modeling assumptions the causal direction between two time series is identifiable. They also show robustness results to downsampling. I really like the work and thank the authors for their submission. I have several minor comments and clarification questions below and would be happy to hear authors' comments on them.

Detailed Comments:
I would explicitly mention that the proposed framework assumes a linear relation between the time series' in the Introduction.

Referring to a classical textbook in signal processing might help with the background exposition such as Fourier transform, e.g., Signals and Systems by Oppenheim and Willsky.

"If such a BIBO-stable filtering relationship exists in only one direction (i.e. when the frequency response is not invertible at some frequency), it is natural to assign causality to the stable direction."
Could you please elaborate on this? I believe the reasoning relies on the assumption that the true model satisfies this property.

BIBO stability seems to be a central concept here, as it appears in Assumption 1. It might be better to provide the definition in a Definition environment rather than in-text for clarity.

"This suggests that the amplification implemented by the mechanism is not adapted to the specific values..."
This statement might be misunderstood. I suggest rephrasing as ".. amplification implemented by the mechanism is not dependent on the specific values..."

"white noise"
Please define white noise in relation to DTFT.

"the ICM-based Trace Condition (Janzing et al., 2010b) has been shown to be equivalent to"
Could you also state the condition?

I find the notation \vec{P_Y} slightly confusing since it is used together with the vector notation. Pushforward measure is typically shown with \mu in the subscript. Authors could consider using this notation instead.

Why is the distance between \vec{P_Y} and U_Y called "irregularity of mechanism" in Figure 1 and last paragraph before Section 4.1? Could you provide some intuition and reasoning for this statement?

Could you comment on why the deterministic mechanism (Y=f(X)) assumption is necessary in Section 4.1 but not for Gaussian processes in Section 4.3? I don't think this is necessary based on the proof; but some insight on what makes it possible that it wasn't before while working with distributions would be appreciated.

A cross-citation is missing from page 17.

Please define "radius distribution".

"U and B are stochastically independent"
Could you clarify this statement? They don't seem to be independent in the probabilistic sence since B=RU so I believe I am missing something here.

guaranties->guarantees

as number of filter coefficients->as the number of filter coefficients

instead m inside the exponential->instead of m inside the exponential

as there product->as their product

If->if in Proposition 2. Please go over Propositon 2 in general. There are several typos.

Could you comment on the assumption between \hat{h} and \textbf{h} in Propositon 2?

Section 5.3 and Propositon 2: It is true that \eta>0 but is there a way to comment on the gap? What do you intuitively expect this gap to depend on? Furthermore, this gap will not be available at the time of inference, correct?

guaranty->guarantee

On key issue->One key issue

Theorem 3 is very nice to have.

It is also very good to see that the method has some robustness against down-sampling.

"where signals which such invariance"->"where signals with such invariance"

**Summary:**

Official Review

---

> ### Author Response · Authors · 2021-12-04
> **Reply to reviewer P6NT**
>
> Thank you for appreciating the value of our contribution! Please find your comments addressed below.
>
> *”I would explicitly mention that the proposed framework assumes a linear relation between the time series' in the Introduction.”*
>
> This is a good point, we will also specify it in the abstract as well.
>
> *”Referring to a classical textbook in signal processing might help with the background exposition such as Fourier transform, e.g., Signals and Systems by Oppenheim and Willsky.”*
>
> We will add this reference in the background section.
>
> *”"If such a BIBO-stable filtering relationship exists in only one direction (i.e. when the frequency response is not invertible at some frequency), it is natural to assign causality to the stable direction."
> Could you please elaborate on this? I believe the reasoning relies on the assumption that the true model satisfies this property.”*
>
> Indeed, if the system were not to be stable, the system would leave its domain of linearity as the output magnitude grows exponentially and the overall assumptions of this paper would be violated. We will mention it explicitly in the background section.
>
> *”BIBO stability seems to be a central concept here, as it appears in Assumption 1. It might be better to provide the definition in a Definition environment rather than in-text for clarity.”*
>
> We will do it.
>
> *“ "This suggests that the amplification implemented by the mechanism is not adapted to the specific values..."
> This statement might be misunderstood. I suggest rephrasing as ".. amplification implemented by the mechanism is not dependent on the specific values..." ”*
>
> Thank you, we will rephrase accordingly.
>
> *”Please define white noise in relation to DTFT.”*
>
> We have defined it at the top of page 5. We can make this definition more formal but please let us know if we misunderstood your comment.
>
> *”
> "the ICM-based Trace Condition (Janzing et al., 2010b) has been shown to be equivalent to"
> Could you also state the condition? ”*
>
> Yes we can restate it formally in the paper (in main text if space constraints allow, or in supplemental). In short, the trace condition postulated that the trace of the effect covariance factorizes as the production of the trace of the cause covariance and another trace factor function of the linear operator linking cause to effect.
>
> *“ I find the notation \vec{P_Y} slightly confusing since it is used together with the vector notation. Pushforward measure is typically shown with \mu in the subscript. Authors could consider using this notation instead.”*
>
> This is a good idea, we will reformulate based on a pushforward notation to avoid confusion with the vector notation.
>
> *” Why is the distance between \vec{P_Y} and U_Y called "irregularity of mechanism" in Figure 1 and last paragraph before Section 4.1? Could you provide some intuition and reasoning for this statement? ”*
>
> Irregularity stems from the fact that it is define as the distance between a point and a manifold of so-called “regular” distributions. In the SIC case, regular distributions are white noises.
>
> *“Could you comment on why the deterministic mechanism (Y=f(X)) assumption is necessary in Section 4.1 but not for Gaussian processes in Section 4.3? I don't think this is necessary based on the proof; but some insight on what makes it possible that it wasn't before while working with distributions would be appreciated.”*
>
> The deterministic mechanism assumption is assumed through all the paper, due to the filtering relation specified in eq. (1). Proofs also rely on this assumption. We can further emphasize this in the background and introduction. Extensions of theoretical analyses of ICM criteria to the noisy mechanism case are actually very challenging.
>
> *”Please define "radius distribution".”*
>
> We actually meant a radius random variable. We will modify it accordingly.
> *” "U and B are stochastically independent"
> Could you clarify this statement? They don't seem to be independent in the probabilistic sence since B=RU so I believe I am missing something here. ”*
>
> This is a synonym for statistical independence (we will replace by this term), as opposed to independence of mechanism. But confusion comes from a typo: we meant $\boldsymbol{U}$ and $R$ are statistically independent. We fixed it.
>
> *“Could you comment on the assumption between \hat{h} and \textbf{h} in Propositon 2?”*
>
> Due to Parseval equality, the squared norm of $\textbf{h}$ also corresponds to the average of $\hat{h}$ in the frequency domain, such that the assumption essentially states that the absolute squared Fourier transform remains below twice its average.
>
> *“Section 5.3 and Propositon 2: It is true that \eta>0 but is there a way to comment on the gap? [...] Furthermore, this gap will not be available at the time of inference, correct?”*
>
> We agree this gap is challenging to obtain, and this has been a difficulty to prove identifiability. At a high level, theorem 3 relies on properties of the generative model to assess this gap.

---

### Official Review · Reviewer_Rp12 · 2021-11-23

**Confidence:** 2
**Overall Score:** 5

**Main Review:**

To be honest, I am not familiar with the field. Therefore, it is hard for me to comment on the contribution and significance of the paper. For example, is SIC an important state-of-the-art method in causal direction inference? How useful is the information geometric interpretation? How broad is the generative model framework?

Regarding the technical details, I have the following questions:

1. Since confounding issues may present under the model in equation (1), I wonder when the authors discuss the causal mechanism using equation (1), do they assume that the causal relationship between $\boldsymbol{x}$ and $\boldsymbol{y}$ has been confirmed, and the goal is only to determine the causal direction between $\boldsymbol{x}$ and $\boldsymbol{y}$? If so, please specify that in the paper.
2. How can we generalise the definite of $\overrightarrow{P}_Y$ below the first equation on page 6 to the notations $\overrightarrow{P_YU_Y}$, $\overrightarrow{P_Y\overrightarrow{P}_Y}$ and $\overrightarrow{\overrightarrow{P}_YU_Y}$? Could the authors be more specific on this?
3. In section 5.1, $\boldsymbol{b}$ is assumed to be sampled from a random variable $\boldsymbol{B}$, where $\boldsymbol{B}=R\boldsymbol{U}$, with $R\geq 0$ a real valued radius distribution and $\boldsymbol{U}$ uniformly distributed on the unit sphere in $\mathbb{R}^m$. Here, what does $R$ really mean, a distribution function or a random variable?

In terms of paper writing, there seems a lot of typos in the main text and Latex compiling errors in the appendix. For example,

1. In the equation at the end of page 3, do the authors actually mean $\hat{\boldsymbol{h}}_{\boldsymbol{Y} \rightarrow \boldsymbol{X}} $ in the denominator on the right-hand side?

2. In the fourth line on page 5, do the authors mean ''define $\rho_{\boldsymbol{Y}\rightarrow\boldsymbol{X}}$ by exchanging the roles of $\boldsymbol{X}$ and $\boldsymbol{Y}$''?
3. In the fourth line of section 5.4, it should be ''One key issue'' instead of ''On key issue''.
4. Latex compiling errors can be found in Appendix A.2 and A.9.
5. Many notations are not consistently presented in the paper. For example, the impulse response is sometimes presented in bold as $\boldsymbol{h}$ and sometimes as an ordinary $h$.

**Summary:**

The paper provides an information geometric interpretation for the Spectral Independence Criterion (SIC) based on the connection between SIC and the Trace Method and that between the Trace Method and Information Geometric Causal  Inference established in the literature. After that, the authors present some identifiability results in a toy generative model and show the robustness of SIC to downsampling. Finally, the paper discusses a way to generalise the spectral independence assumption through the invariance perspective.

---

> ### Author Response · Authors · 2021-12-04
> **Reply to Reviewer Rp12**
>
> Thank you for your comments that we address below.
> *“Since confounding issues may present under the model in equation (1), I wonder when the authors discuss the causal mechanism using equation (1), do they assume that the causal relationship has been confirmed, and the goal is only to determine the causal direction? “*
>
> Yes, only determining the causal direction is addressed. We will specify more clearly the cause-effect pair inference setting in section 2.
>
> *”How can we generalise the definition of $\overrightarrow{P_Y}$
> below the first equation on page 6 to the notations $\overrightarrow{P_Y U_Y}$? Could the authors be more specific on this?”*
>
> These last notations above equation (7) are generalized vector notations for oriented pairs of points (each point being a probability distribution) in the statistical manifold. The squared norm of these vectors corresponds to the KL divergence, and a generalized form of orthogonality between is specified by generalized Pythagorean theorems in the form of the formula at the top of page 6. These vector representations are illustrated on Figure 1.
> We will clarify this in section 4.1 and change the notation for  $\overrightarrow{P_Y}$ as suggested by another reviewer.
>
> *”What does R really mean, a distribution function or a random variable?”*
>
> We mean a random variable, we will fix it, thanks for pointing that out.
>
> *”In the equation at the end of page 3, do the authors actually mean $\boldsymbol{h}_{Y\rightarrow X}$ in the denominator on the right-hand side?”*
>
> Yes we fixed this typo, thank you.
>
> *“In the fourth line on page 5, do the authors mean ''define
> by exchanging the roles of and ''?”*
>
> Yes, we fixed this typo.
>
> *“In the fourth line of section 5.4, it should be ''One key issue'' instead of ''On key issue''.”*
>
> Thank you. this is corrected.
>
> *“Latex compiling errors can be found in Appendix A.2 and A.9.”*
>
> Thank you we will fix them.
>
> *”Many notations are not consistently presented in the paper. For example, the impulse response is sometimes presented in bold as and sometimes as an ordinary”*
>
> We denoted the impulse response sequence in bold and its Fourier transform (with a hat) in ordinary font (as it is a function of a real variable). We will do a careful pass to check the consistency.

---

### Official Review · Reviewer_SghD · 2021-11-24

**Confidence:** 4
**Overall Score:** 5

**Main Review:**

The paper is well written, an easy read.
The problem is quite interesting to the community and has plenty of applications in different areas. Exploring the idea of SIC and establishing its information geometric interpretation is valuable.

SIC seems to be a strong assumption which limits the applicability of this method. Is SIC verifiable from observed data? A discussion on what classes of dynamical systems satisfy this assumption would improve the acceptability of SIC.

My main concern is about the contribution and novelty. Comparing the result with [Shajarisales et al. (2015)], this work offers small amount of contribution (e.g., information geometric interpretation and Robustness with respect to Decimation procedure).

Generalizability of this approach is questionable. Consider a simple scenario in which X->Z->Y. In this case, does SIC-based approach detect a direct influence from X to Y? and how does this approach take information of time series Z into account for detecting causal-effects?
What if there are hidden confounder between X and Y? Can one still apply SIC approach?

In Theorem 2, \epsilon cannot be arbitrary small. If it goes to zero, then the probability becomes -1. Hence, given that <S_{xx}> is related to m and m^3\epsilon^2 cannot be arbitrary small, is the proposed bound in this theorem a tight bound?

The experiments are not convincing that this method can be promising. Is it better than other approaches for identifying cause vs effect e.g., the Wiener filter approach in “On the Problem of Reconstructing an Unknown Topology via Locality Properties of the Wiener Filter,” by Donatello Materassi et al. 2012. This is yet another concern that the cited papers are not diverse (limited  and seems to ignore other literature in causal discovery in time series using PSD.

Minor comments:
Equation after Eq. (2) is incorrect.
There are several missing references in the Appendix.

**Summary:**

This paper studies estimating the directionality of cause-effect relationships between two time series using the principle of Independence of Causal Mechanisms (ICM). ICM leads to the Spectral Independence Criterion (SIC), postulating that the power spectral density (PSD) of the cause time series is uncorrelated with the squared modulus of the frequency response of the filter generating the effect. This work provids an information theoretic interpretation of SIC, presents an identifiability result, and demonstrates the robustness of SIC to downsampling.

---

> ### Author Response · Authors · 2021-12-04
> **Reply to Reviewer SghD**
>
> Thank you for your comments. We address them below.
>
> *“SIC seems to be a strong assumption which limits the applicability of this method. Is SIC verifiable from observed data?”*
>
> As described in sec. 7, we can check that the distribution of SDR values is centered around one in empirical data. A deviation from this value suggests that SIC assumptions are not met and need to be adapted. See illustration on Fig. 4 c-d for power law like neural signals.
>
> *“A discussion on what classes of dynamical systems satisfy this assumption would improve the acceptability of SIC.”*
>
> As stated in the background section, our theory and the original paper rely on linearity, stationarity and deterministic mechanism assumptions. We will emphasize it more in the introduction and background.
>
> *“My main concern is about the contribution and novelty. Comparing the result with [Shajarisales et al. (2015)], this work offers small amount of contribution (e.g., information geometric interpretation and Robustness with respect to Decimation procedure).”*
>
> In addition to these two results, we offer for the first time an identifiability result, which is a (if the not “the”) central goal of theories of causal inference. In addition, sec. 7 provides an extension of this setting which is highly relevant in applications to experimental data (as they often comprise power law signals).
>
> *“Generalizability of this approach is questionable. Consider a simple scenario in which X->Z->Y. In this case, does SIC-based approach detect a direct influence from X to Y? and how does this approach take information of time series Z into account for detecting causal-effects? What if there are hidden confounder between X and Y? Can one still apply SIC approach?”*
>
> Confounding is a very interesting question but out of the scope of the present work, we will emphasize this in the background section (this is also mentioned in the discussion). The present paper focuses on inferring the direction of causation for cause-effect pairs and can be applied to unconfounded pairs of nodes satisfying our assumptions. In the case that you mention, we would thus test X->Z, Z->Y or even X->Y, depending on the question asked. Some further aspects of the case of X->Z->Y have been investigated empirically in Besserve et al. 2021.
>
> *“In Theorem 2, \epsilon cannot be arbitrary small. If it goes to zero, then the probability becomes -1. Hence, given that <S_{xx}> is related to m and m^3\epsilon^2 cannot be arbitrary small, is the proposed bound in this theorem a tight bound?”*
>
> We did not perform any analysis of the tightness of the bound and are not aware of concentration of measure results which provide such guaranties. Note that tightness of the bound is not required to achieve our purpose: identifiability.
>
> *“The experiments are not convincing that this method can be promising. Is it better than other approaches for identifying cause vs effect e.g., the Wiener filter approach in “On the Problem of Reconstructing an Unknown Topology via Locality Properties of the Wiener Filter,” by Donatello Materassi et al. 2012. ”*
>
> First let us mention that the paper by Donatello et al. 2021 looks like a very interesting theoretical work, but does not provide quantitative comparisons with classical baselines, nor provides a link to a code to test it. Please refer to Shajarisales et al. (2015) for comparison of SIC with competing baselines like Granger causality. In the present work, we focus instead on better understanding the theoretical foundations of this approach. On a high level, while Granger-like approaches lend themselves to fairly classical identifiability analysis of auto-regressive models, this paper and Shajarisales et al. (2015) suggest SIC can be useful under a set of misspecifications of this model.
>
> *" This is yet another concern that the cited papers are not diverse (limited and seems to ignore other literature in causal discovery in time series using PSD."*
> We are happy to include a greater variety of references, although we are not aware of other methods relying on PSD in the strict sense. However, we can include references to partial directed coherence and other frequency domain approach, exploiting cross spectral densities.
>
> *“Minor comments: Equation after Eq. (2) is incorrect. There are several missing references in the Appendix.”*
>
> Thank you we fixed these typos.

---

### Decision · Program_Chairs · 2022-01-13

**Decision:**

Accept (Poster)

**Comment:**

The paper provides theoretical foundations for the spectral independence criterion which states power spectral density of the cause time series is uncorrelated with the squared modulus of the frequency response of the filter generating the effect. The authors introduce an information geometric interpretation, provide an identifiability result and study the robustness of the criterion with respect to downsampling. Overall, the results in the paper are a modest but useful step from the results in Shajarisales et al. (2015). That being said, I found the paper hard to read. Since the overall framework is relatively far from standard causal frameworks, the paper would benefit from some additional explanations (for example, when stating Postulate 1).